# Activation of actin-depolymerizing factor by CDPK16-mediated phosphorylation promotes actin turnover in *Arabidopsis* pollen tubes

Qiannan Wang[1], Yanan Xu[1], Shuangshuang Zhao[2,3], Yuxiang Jiang[1], Ran Yi[1], Yan Guo[2], Shanjin Huang[1]*

1 Center for Plant Biology, School of Life Sciences, Tsinghua University, Beijing, China, 2 State Key Laboratory of Plant Physiology and Biochemistry, College of Biological Sciences, China Agricultural University, Beijing, China, 3 Key Laboratory of Plant Stress, Life Science College, Shandong Normal University, Jinan, China

☯ These authors contributed equally to this work.
* sjhuang@tsinghua.edu.cn

**Data Availability Statement:** All relevant data are within the paper and its Supporting Information files.

## Abstract

As the stimulus-responsive mediator of actin dynamics, actin-depolymerizing factor (ADF)/cofilin is subject to tight regulation. It is well known that kinase-mediated phosphorylation inactivates ADF/cofilin. Here, however, we found that the activity of *Arabidopsis* ADF7 is enhanced by CDPK16-mediated phosphorylation. We found that CDPK16 interacts with ADF7 both *in vitro* and *in vivo*, and it enhances ADF7-mediated actin depolymerization and severing *in vitro* in a calcium-dependent manner. Accordingly, the rate of actin turnover is reduced in *cdpk16* pollen and the amount of actin filaments increases significantly at the tip of *cdpk16* pollen tubes. CDPK16 phosphorylates ADF7 at Serine128 both *in vitro* and *in vivo*, and the phospho-mimetic mutant ADF7$^{S128D}$ has enhanced actin-depolymerizing activity compared to ADF7. Strikingly, we found that failure in the phosphorylation of ADF7 at Ser128 impairs its function in promoting actin turnover *in vivo*, which suggests that this phospho-regulation mechanism is biologically significant. Thus, we reveal that CDPK16-mediated phosphorylation up-regulates ADF7 to promote actin turnover in pollen.

## Introduction

The actin cytoskeleton is a dynamic and signaling-responsive structure that has been implicated in numerous physiological cellular processes including cytokinesis, cell migration, polarized cell growth, and various intracellular trafficking events [1]. A dynamic actin cytoskeleton is absolutely required for polarized growth of pollen tubes [2,3], which provide passage for 2 non-motile sperm cells to ensure double fertilization in flowering plants. Previous studies revealed that there exists a distinct population of extremely dynamic actin filaments in pollen tubes [2–5]. These filaments assume a distinct spatial distribution and form a unique structure called the "apical actin structure" at pollen tube tips [6,7]. It is fascinating to consider how pollen tubes control the dynamics and spatial distribution of apical actin filaments in response to various signals during the rapid extension of pollen tubes.

**Funding:** This work was supported by the National Key R&D Program of China (2022YFA1303400 to S.H. and Y.G.) and the National Natural Science Foundation of China (32270338 and 31970180 to S.H.). The funders had no role in study design, data collection and analysis, decision to publish, or preparation of the manuscript.

**Competing interests:** The authors have declared that no competing interests exist.

**Abbreviations:** ABP, actin-binding protein; ADF, actin-depolymerizing factor; CDPK, calcium-dependent protein kinase; LatB, latrunculin B; LCI, luciferase complementation imaging; LC-MS/MS, liquid chromatography–mass spectrometry/mass spectrometry; PM, plasma membrane; qRT-PCR, quantitative real-time PCR; TIRFM, total internal reflection fluorescence microscopy; WT, wild type.

Actin-depolymerizing factor (ADF)/cofilin, a central regulator of actin dynamics, is a stimulus-responsive protein [8] which has been implicated in the regulation of actin dynamics in pollen tubes [9–13]. The activity of ADF/cofilin is subject to tight regulation by kinase-mediated phosphorylation [14–17], pH [18–22], and interactions with other binding partners, such as CAP1 [23,24], AIP1 [25–27], and Coronin [28], etc. Kinase-mediated phosphorylation represents a well-known inactivation mechanism of ADF/cofilin, with the phosphorylation occurring at Ser3 or Ser6 of ADFs/cofilins from different organisms [8,29]. This type of phospho-regulation has been reported for plant ADFs [30,31]. Specifically, plant ADFs were demonstrated or proposed to be inactivated by calcium ($Ca^{2+}$)-dependent protein kinase (CDPK)-mediated phosphorylation [29,31,32], but the biological significance of this regulatory mechanism remains to be explored. In particular, it remains a mystery how ADF contributes to the enhancement of actin dynamics at pollen tube tips where CDPK(s) is supposed to be active.

Here, we report that CDPK16 up-regulates the activity of pollen-specific *Arabidopsis* ADF7 by phosphorylating it at Ser128. Our finding differs from the consensus view that kinase-mediated phosphorylation inactivates ADF/cofilin. Our study thus significantly enhances our understanding of the regulation of ADF/cofilin. In addition, our findings provide the direct evidence linking $Ca^{2+}$/CDPK signaling to actin dynamics during pollen tube growth and suggest a scenario in which high $Ca^{2+}$-CDPK activity can be unified with ADF7 activation to maintain the high dynamics of actin filaments at pollen tube tips. Our study thus sheds light on the functional adaptation of ADF/cofilin and the regulation of actin dynamics during polarized pollen tube growth.

## Results

### Loss of function of *CDPK16* reduces the rate of actin turnover in pollen

To understand the regulation of actin turnover in pollen, we performed forward chemical genetic screening to uncover mutations that alter the sensitivity of pollen germination to latrunculin B (LatB). Our attention was attracted by 1 T-DNA insertion knockout mutant allele of *CDPK16*, designated as *cdpk16-1* (S1A and S1B Fig). Germination of *cdpk16-1* pollen is resistant to LatB (see below). To demonstrate that the LatB-resistant pollen germination phenotype is indeed caused by the mutation in *CDPK16*, we also generated another *cdpk16* mutant allele by the CRISPR/cas9 approach [33], designated as *cdpk16-2* (S1C Fig). We found that *cdpk16-1* and *cdpk16-2* mutant pollen germinates better than wild-type (WT) pollen in the presence of LatB (S1D and S1E Fig), which suggests that the germination of *cdpk16* mutant pollen is resistant to LatB. Accordingly, we found that the relative growth rate of pollen tubes was increased significantly in *cdpk16* mutants compared to WT in the presence of LatB (S1F and S1G Fig), which suggests that loss of function of *CDPK16* renders pollen tube growth resistant to LatB. We next found that there is no overt difference in terms of the overall brightness and organization of actin filaments in *cdpk16* mutant pollen grains compared to WT in the absence of LatB (S2A–S2C Fig). We noticed that actin filaments became fragmented in both WT and *cdpk16* mutant pollen grains after treatment with 150 nM LatB, but the extent of fragmentation is less obvious in *cdpk16* mutant pollen grains than in WT (S2A Fig). In addition, we found that LatB-triggered actin depolymerization is inhibited in *cdpk16* mutant pollen grains compared to WT, as evidenced by the significantly higher relative amount of actin filaments in *cdpk16* mutant pollen grains compared to WT (S2B and S2C Fig). In line with the above observations, we found that overexpression of *CDPK16* (S3A Fig) renders pollen germination sensitive to LatB (S3B–S3D Fig). Thus, these data suggest that CDPK16 promotes actin turnover in pollen.

## Loss of function of *CDPK16* promotes pollen germination and inhibits pollen tube growth

We next determined the role of CDPK16 in regulating pollen germination and pollen tube growth. We initially compared the time course of pollen germination in WT and the 2 *cdpk16* mutants, and found that pollen germination is accelerated early on in the *cdpk16* mutants, but the overall pollen germination rate in *cdpk16* mutants does not differ from that in WT (S4A Fig). This suggests that loss of function of *CDPK16* promotes pollen germination. In addition, we found that the rate of pollen tube growth is significantly reduced in *cdpk16* mutants compared to WT (S4B and S4C Fig), which suggests that *CDPK16* promotes normal pollen tube growth. These data together suggest that CDPK16 maintains the normal rate of pollen germination and promotes pollen tube growth.

## Loss of function of *CDPK16* induces obvious accumulation of actin filaments at pollen tube tips

Next, we performed real-time visualization of the dynamics of actin filaments decorated with Lifeact-eGFP and found that, consistent with previous observations [5], actin filaments continuously polymerized from the plasma membrane (PM) within the apical region of pollen tubes (Fig 1A and S1 Movie). We found that PM-originated actin filaments are obviously brighter within the apical region of *cdpk16* pollen tubes than in WT pollen tubes (Fig 1A and S2 Movie). Kymograph analysis showed that the region occupied by membrane-originated actin filaments was enlarged in *cdpk16-1* pollen tubes compared to WT (Fig 1A, right panel). This is supported by physical measurements showing that the distance from the base of the apical actin structure to the pollen tube tip (mean ± SE) increased significantly in *cdpk16-1* pollen tubes (6.20 ± 0.047 μm) compared to WT pollen tubes (4.27 ± 0.037 μm) (Fig 1B). Through monitoring the dynamics of individual PM-originated actin filaments (Fig 1C and S3 and S4 Movies), we found that the average severing frequency of actin filaments was significantly reduced in *cdpk16-1* pollen tubes compared to WT (Fig 1D). Accordingly, the maximal filament length and lifetime increased significantly in *cdpk16-1* pollen tubes compared to WT (Fig 1D). However, we did not notice obvious differences in the elongation and depolymerization rates of PM-originated apical actin filaments in *cdpk16-1* pollen tubes compared to WT (Fig 1D). Thus, our results suggest that the amount of apical actin filaments is increased and their dynamics are reduced in *cdpk16-1* pollen tubes.

## CDPK16 interacts with and phosphorylates ADF7 both *in vitro* and *in vivo* and it enhances the actin-depolymerizing and severing activities of ADF7 *in vitro*

To determine whether CDPK16 regulates actin turnover via interaction with certain actin-binding proteins (ABPs) *in vivo*, we performed pull-down experiments followed by mass spectrometry to search for candidate interacting proteins of CDPK16. Interestingly, we found that ADF7 is enriched in the CDPK16 pull-down fraction (Fig 2A–2C). The direct interaction between CDPK16 and ADF7 was confirmed by the luciferase complementation imaging (LCI) assay (Fig 2D) and further validated by showing that CDPK16 can phosphorylate ADF7 *in vitro* (Fig 2E). To determine whether CDPK16 can phosphorylate ADF7 *in vivo*, we decided to treat total pollen proteins with phosphatase in the presence and absence of CDPK16, followed by 2D electrophoresis and antibody detection of ADF7. However, the currently available anti-ADF7 antibody cross-reacts with the highly similar ADF10, so we initially analyzed *adf7* and *adf10* mutants to distinguish ADF7 from ADF10 after electrophoresis (Fig 2F). Comparing the

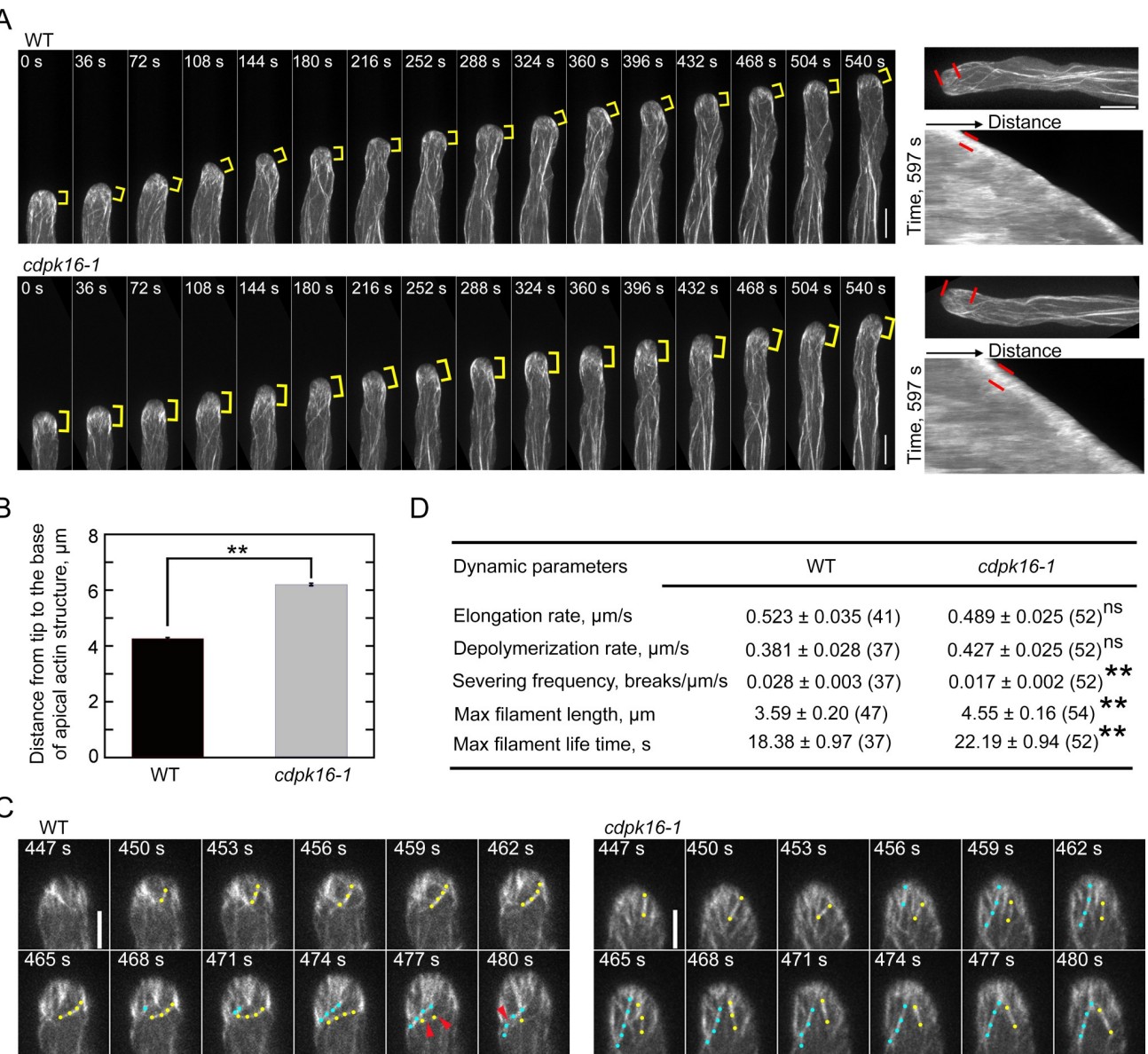

**Fig 1. Actin dynamics is reduced in *cdpk16* pollen tubes.** (**A**) Time-lapse images of actin filaments decorated with Lifeact-eGFP in growing WT and *cdpk16-1* pollen tubes. Yellow brackets indicate the region occupied by membrane-originated actin filaments. The right panels are Kymograph analyses of the growing WT and *cdpk16-1* pollen tubes shown in the left panel. The region occupied by the membrane-originated actin filaments is marked by 2 red lines. Bar = 10 μm. (**B**) Quantification of the width of the region occupied by membrane-originated actin filaments in WT and *cdpk16-1* pollen tubes. Data are presented as mean ± SE, \*\*$P < 0.01$ (Student's *t* test). More than 400 time points from 10 pollen tubes were measured. Numerical data underlying this panel are available in S1 Data. (**C**) Time-lapse images of actin filaments at the apical region in WT and *cdpk16-1* pollen tubes. Apical actin filaments are indicated by different colored dots. The severing events of actin filaments are indicated by red arrows. Bar = 5 μm. (**D**) Dynamic parameters of apical actin filaments in WT and *cdpk16-1* pollen tubes. Data are presented as mean ± SE, with the number of filaments in parentheses. NS, no significant difference, \*\*$P < 0.01$, (Student's *t* test). WT, wild type.

results from WT, *adf7* and *adf10*, it appears that ADF7 might be subject to posttranslational modification, as there are several protein spots corresponding to ADF7 (Fig 2F). Protein spot (a) is the presumed phosphorylated form of ADF7, based on its mobility after 2D electrophoresis (Fig 2F). In support of this speculation, we found that treatment with λ-phosphatase reduced the intensity of spot (a) in total proteins extracted from WT pollen (Fig 2G).

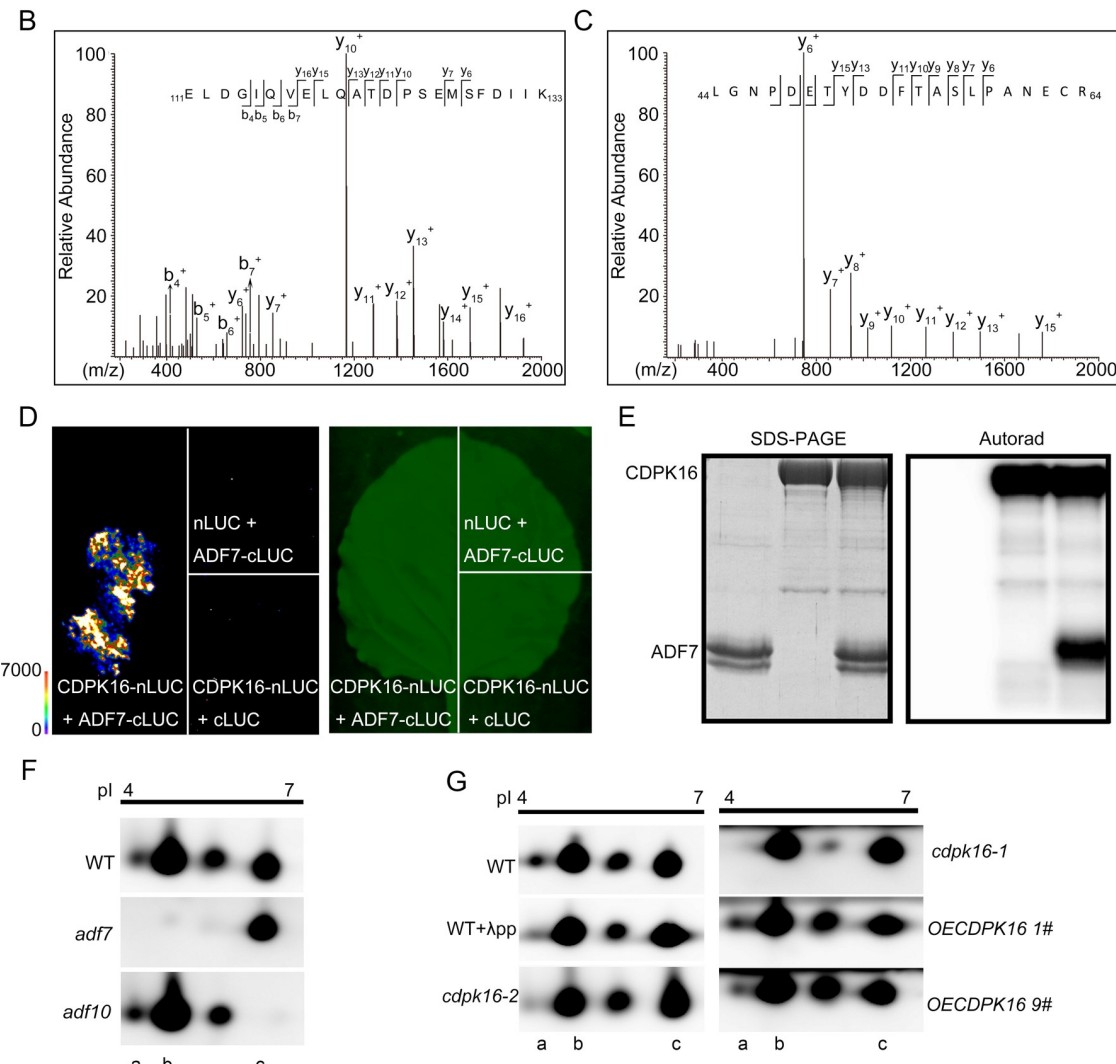

**A** Identification table

| Protein name | Gene ID | Unique peptide | Sequence coverage |
|---|---|---|---|
| CPK28 | At5g66210 | 2 | 17.01% |
| Mucin-like protein | At3g28830 | 4 | 14.10% |
| GLOX1 | At1g67290 | 12 | 30.73% |
| GRP14 | At5g07510 | 3 | 23.20% |
| BGAL11 | At4g35010 | 7 | 14.44% |
| SKS12 | At1g55570 | 5 | 20.00% |
| ADF7 | At4g25590 | 2 | 32.12% |
| Pectate lyase family protein | At5g15110 | 3 | 11.23% |
| … | … | … | … |

**Fig 2. CDPK16 interacts with and phosphorylates ADF7.** (**A**) Identification of proteins in CDPK16-6×His pull-down fractions by LC-MS/MS. ADF7 is one of the proteins that appears in the pull-down fraction. (**B**, **C**) Two representative peptides derived from ADF7 are presented. A full list of the peptides is presented in S2 Data. (**D**) CDPK16 interacts with ADF7. The interaction between ADF7 and CDPK16 was determined by qualitative analysis of luciferase (LUC) activity using the LCI assay. (**E**) CDPK16 phosphorylates ADF7 *in vitro*. The left panel shows the CDPK16-6×His and ADF7 input proteins, which were detected by Coomassie Brilliant blue R 250 staining. The right panel shows phosphorylated ADF7, which was detected by [γ-$^{32}$P] ATP autoradiography. The original pictures are available in S1 Raw Images. (**F**) Detection of ADF7 and ADF10 protein spots by 2D gel-electrophoresis. Total proteins from WT, *adf7* and *adf10* pollen were separated by 2D gel-electrophoresis. Protein spots were revealed by western blot analysis probed with anti-ADF7 antibody, which also detects ADF10. The original pictures are available in S1 Raw Images. (**G**) Detection of ADF7 and ADF10 in total proteins extracted from WT and *cdpk16* pollen. Total proteins from WT pollen in the presence and absence of λpp or from pollen of *cdpk16* mutants and *CDPK16* overexpressors were separated by 2D gel-electrophoresis and subjected to western blot analysis probed with anti-ADF7 antibody. (a), (b), and (c) in (**F**) and (**G**) represent phosphorylated ADF7, ADF7, and ADF10, respectively. The original pictures are available in S1 Raw Images. ADF, actin-

depolymerizing factor; LCI, luciferase complementation imaging; LC-MS/MS, liquid chromatography–mass spectrometry/mass spectrometry; WT, wild type.

Furthermore, we found that the intensity of protein spot (a) is reduced in *cdpk16* mutant pollen total extract whereas is increased in pollen total extract from *CDPK16* overexpressors compared to WT (Fig 2G), which suggests that CDPK16 can phosphorylate ADF7 in pollen.

Next, using a high-speed F-actin co-sedimentation assay, we found that CDPK16 increased the activity of ADF7 in depolymerizing actin filaments *in vitro* (Fig 3A and 3B), and this enhancement by CDPK16 was reduced in the absence of $Ca^{2+}$ (Fig 3C and 3D). These results suggest that CDPK16 promotes the actin-depolymerizing activity of ADF7 in a $Ca^{2+}$-dependent manner. We previously showed that the actin disassembly activity of ADF4 is regulated by phosphorylation in stomata [30]. Interestingly, we found that CDPK16 only weakly, albeit significantly, enhanced the actin-depolymerizing activity of ADF4 (S5 Fig), which suggests that CDPK16 promotes the actin-depolymerizing activity of ADF7 in a somewhat specific manner. In addition, we found that CDPK16 enhanced the activity of ADF7 in shortening actin filaments *in vitro* (Fig 3E and 3F). As ADF/cofilin also contributes to actin depolymerization via fragmentation of actin filaments [34] and ADF7 can sever actin filaments [10], we determined whether CDPK16 can promote the severing activity of ADF7. We used total internal reflection fluorescence microscopy (TIRFM) to show that CDPK16 enhanced the activity of ADF7 in severing actin filaments (Fig 3G and 3H and S5–S8 Movies). Thus, our study suggests that CDPK16 enhances ADF7-mediated actin depolymerization and severing *in vitro*.

### Overexpression of *ADF7* alleviates the LatB-resistant pollen germination phenotype in *cdpk16* and loss of function of *CDPK16* enhances the LatB-resistant pollen germination phenotype in *adf10*

To determine whether CDPK16 promotes actin turnover through activating ADF7, we tested whether gain of function of *ADF7* can alleviate the actin turnover defects in *cdpk16* mutant pollen. Indeed, we found that overexpression of *ADF7* suppressed the actin turnover phenotype in *cdpk16* mutant pollen (S6A and S6B Fig). ADF7 and ADF10 act redundantly to control actin turnover in pollen [11], and our results above suggest that ADF7 might be the more relevant substrate of CDPK16 *in vivo* (Fig 2F and 2G). Therefore, we wondered whether loss of function of *CDPK16* will cause an additive effect on actin turnover induced by loss of function of *ADF10* in pollen. As expected, we found that the LatB-resistant pollen germination phenotype is more severe in *adf10 cdpk16-1* double mutants *adf10* single mutants (S6C and S6D Fig). These data together suggest that CDPK16 promotes actin turnover at least partly through up-regulating ADF7 activity in pollen.

### CDPK16 enhances the actin-depolymerizing activity of ADF7 by phosphorylating its Ser128

To uncover the phosphorylation site(s) of ADF7 *in vivo*, we performed mass spectrometry analysis on 8His-ADF7 pulled down from total proteins extracted from pollen. An ADF7 phospho-peptide was repeatedly identified with the phosphate group conjugated to Ser128 (Fig 4A). Strikingly, we found that CDPK16 adds phosphate to the same serine of ADF7 *in vitro* (Fig 4B). This suggests that Ser128 in ADF7 might be the major site of phosphorylation by CDPK16. In support of this notion, we found that the extent of CDPK16-mediated ADF7 phosphorylation *in vitro* was reduced significantly after Ser128 was replaced with aspartic acid (ADF7^S128D) (Fig 4C and 4D). Furthermore, we found that the charge behavior of ADF7^S128D

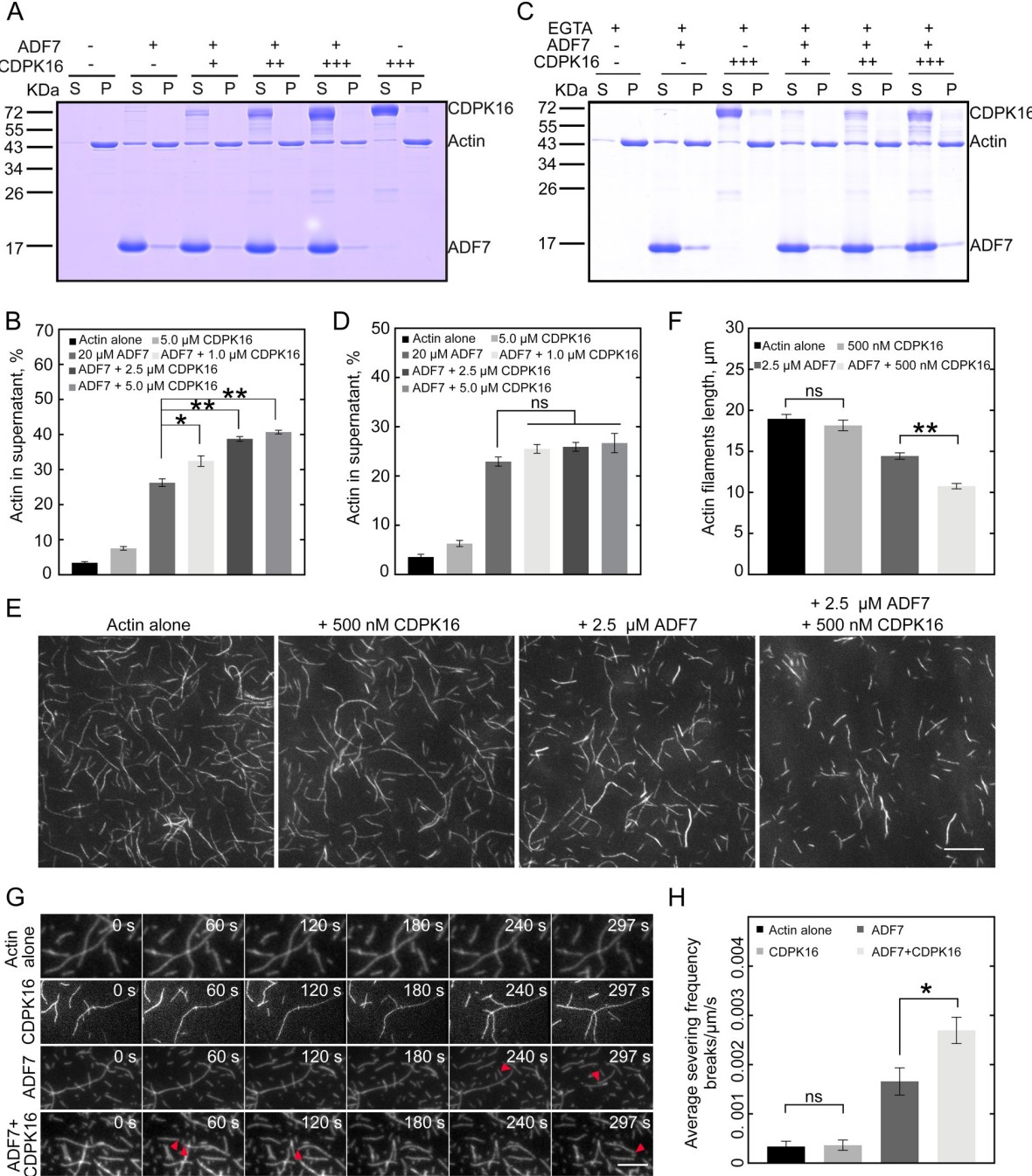

**Fig 3. CDPK16 enhances the activity of ADF7 in a Ca²⁺-dependent manner *in vitro*.** (**A**) SDS-PAGE analysis of the protein samples from a high-speed F-actin co-sedimentation experiment in the presence of $Ca^{2+}$. F-actin, 3 μM; ADF7, 20 μM; CDPK16 (+), 1.0 μM; CDPK16 (++), 2.5 μM; CDPK16 (+++), 5.0 μM. The supernatant fractions (S) and pellets (P) were separated on SDS-PAGE gels, and proteins were detected by Coomassie Brilliant blue R 250 staining. The original pictures are available in S1 Raw Images. (**B**) Quantification of the amount of actin in the supernatant fractions shown in (**A**). Data are presented as mean ± SE, $n = 3$, $^{*}P < 0.05$ and $^{**}P < 0.01$ by Student's $t$ test. Numerical data underlying this panel are available in S3 Data. (**C**) SDS-PAGE analysis of the protein samples from a high-speed F-actin co-sedimentation experiment in the absence of $Ca^{2+}$. The conditions were exactly the same as in (**A**) except for the presence of 0.5 mM EGTA instead of 0.5 mM $CaCl_2$ in the kinase reaction buffer. F-actin, 3 μM; ADF7, 20 μM; CDPK16 (+), 1.0 μM; CDPK16 (++), 2.5 μM; CDPK16 (+++), 5.0 μM. The original pictures are available in S1 Raw Images. (**D**) Quantification of the amount of actin in the supernatant fractions shown in (**C**). Data are presented as mean ± SE, $n = 3$, ns, no significant difference by Student's $t$ test. Numerical data underlying this panel are available in S3 Data. (**E**) Images of actin filaments stained with equimolar Rhodamine-Phalloidin. The concentration of actin filaments is 2 μM. Bar = 10 μm. (**F**) Quantification of the length of actin filaments. Data are presented as mean ± SE, $n = 3$, ns, no significant difference, $^{**}P < 0.01$ by Student's $t$ test. Numerical data underlying this panel are available in S3 Data. (**G**) Time-lapse images of actin filaments. F-

actin, 150 nM (50% Oregon green-labeled); ADF7, 500 nM; CDPK16, 125 nM. The red arrows indicate actin filament severing events. Bar = 10 μm. (**H**) Quantification of the average severing frequency of actin filaments. Data are presented as mean ± SE, ns, no significant difference, *$P < 0.05$ by Student's *t* test. Numerical data underlying this panel are available in S3 Data. ADF, actin-depolymerizing factor; CDPK, calcium-dependent protein kinase.

and non-phosphorylatable ADF7$^{S128A}$ (Ser128 replaced with Alanine) is similar to that of the predicted phosphorylated ADF7 and non-phosphorylated ADF7 (Fig 4E), respectively. Ser128 is conserved in all the class II ADFs in *Arabidopsis*, i.e., ADF7, ADF8, ADF10, and ADF11 (S7 Fig) and in class II ADFs from other plant species (S8 Fig). We next generated a poly-clonal antibody against this phospho-peptide, designated as anti-phospho-ADF7(Ser128), and found that it specifically recognizes CDPK16-phosphorylated ADF7 (S9A Fig). Interestingly, we found that CDPK16 phosphorylates ADF7 in a Ca$^{2+}$-dependent manner (S9B and S9C Fig). Importantly, we found that it recognizes 8His-ADF7 pulled down from total pollen extract, and treatment with λ-phosphatase reduced the signal (S9D Fig), which further suggests that phosphorylation of ADF7 at Ser128 does occur *in vivo*.

We next examined the actin-depolymerizing activity of Ser128 mutants of ADF7. Compared to WT ADF7, ADF7$^{S128D}$ had enhanced actin-depolymerizing activity and ADF7$^{S128A}$ had roughly similar activity, as determined by the high-speed F-actin co-sedimentation assay (Fig 4F and 4G) and the kinetic actin-depolymerizing assay (Fig 4H). This was further confirmed by direct visualization of actin filaments (Fig 4I and 4J). Importantly, we found that CDPK16 failed to enhance the actin-depolymerizing activity of ADF7$^{S128A}$ *in vitro* (S10 Fig), which suggests that Ser128 of ADF7 is the major residue targeted by CDPK16. These data together suggest that Ser128 in ADF7 is phosphorylated by CDPK16, and phosphorylation of this residue increases ADF7 activity.

## ADF7$^{S128A}$ and ADF7$^{S128D}$ fail to fully replace the role of ADF7 in promoting actin turnover *in vivo*

Next, we determined the biological significance of this phospho-regulation mechanism by introducing the non-phosphorylatable ADF7$^{S128A}$ and phospho-mimetic ADF7$^{S128D}$ into pollen. Considering that ADF7 and ADF10 redundantly regulate actin turnover in pollen [11], we assumed that the functional difference between ADF7$^{S128A}$ and ADF7 or ADF7$^{S128D}$ and ADF7 might be amplified in the *adf10* mutant background. We selected transgenic lines containing comparable amounts of ADF7$^{S128A}$, ADF7$^{S128D}$, and ADF7 both transcriptionally (S11A Fig) and translationally (S11B and S11C Fig) for subsequent analyses. We initially found that ADF7$^{S128A}$ functions almost the same as ADF7 in supporting pollen tube growth, whereas ADF7$^{S128D}$ has slightly but significantly higher activity than ADF7 and ADF7$^{S128A}$ in this regard (S11D and S11E Fig). However, in the presence of LatB, we found that pollen tubes harboring both ADF7$^{S128A}$ and ADF7$^{S128D}$ grew significantly faster than pollen tubes harboring ADF7 (S11D and S11F Fig), which suggests that ADF7$^{S128A}$ and ADF7$^{S128D}$ have reduced activity in promoting actin turnover in pollen tubes compared to ADF7. It is quite puzzling that ADF7$^{S128D}$ has reduced capability in promoting actin turnover in pollen compared to ADF7, as ADF7$^{S128D}$ has higher activity in depolymerizing and severing actin filaments than ADF7 *in vitro* (Fig 4F–4J). The reasonable explanation here is that ADF7$^{S128D}$ cannot fully mimic the function of phosphorylated ADF7 in pollen. Strikingly, we found that the germination of pollen harboring ADF7$^{S128A}$ is resistant to LatB compared to pollen harboring WT ADF7 when *CDPK16* is overexpressed (S12 Fig). This suggests that phosphorylation of ADF7 at its Ser128 mainly accounts for the role of CDPK16 in promoting actin turnover in pollen.

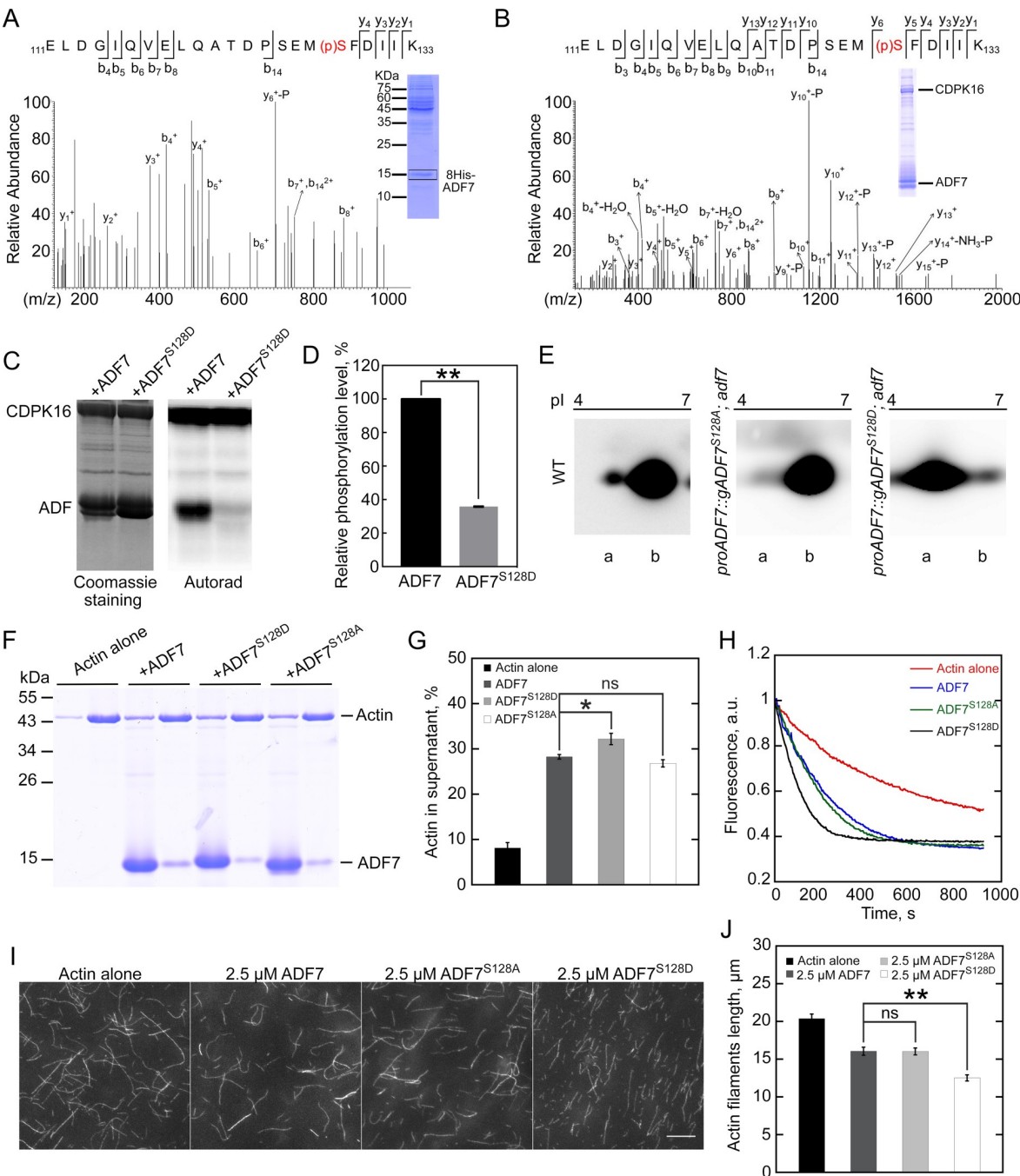

**Fig 4. CDPK16 can phosphorylate Ser128 in ADF7, and ADF7^{S128D} has enhanced actin-depolymerizing and severing activity *in vitro*.**
(**A**) Identification of a phosphorylated peptide with the phosphate group conjugated to Ser128. 8His-ADF7 protein was isolated from pollen grains derived from *proADF7::8His-gADF7; adf7* and subjected to LC-MS analysis. The original LC-MS data underlying this panel are available in S4 Data. The original pictures are available in S1 Raw Images. (**B**) CDPK16 can phosphorylate Ser128 in ADF7 *in vitro*. After incubation of 20 μM ADF7 with 5 μM CDPK16 in kinase buffer for 30 min, the sample was separated by SDS-PAGE. The ADF7 band was cut out and subjected to mass spectrometry analysis. A phosphorylated ADF peptide with the phosphate group conjugated to Ser128 was identified. The original LC-MS data underlying this panel are available in S4 Data. The original pictures are available in S1 Raw Images. (**C**) CDPK16 fails to phosphorylate ADF7^{S128D}-6×His. Purified ADF7-6×His and ADF7^{S128D}-6×His were incubated with CDPK16 in a kinase reaction buffer and the phosphorylation signals were detected by [γ-^{32}P] ATP autoradiography. The original pictures are available in S1 Raw Images. (**D**) Quantification of the relative phosphorylation level of ADF7 shown in (**C**). The data are presented as mean ± SE, $n = 3$, **$P < 0.01$ (Student's *t* test). Numerical data underlying this panel are available in S4 Data. (**E**) 2D electrophoresis assay. Total proteins from mature pollen of WT, *proADF7::gADF7^{S128A}; adf7* and *proADF7::gADF7^{S128D}; adf7* were subjected to 2D electrophoresis analysis. Immunoblotting was performed and probed with anti-ADF7 antibody. (a) Indicates the presumed

phosphorylated ADF7 or ADF7^S128D and (b) indicates unphosphorylated ADF7 or ADF7^S128A. The original pictures are available in S1 Raw Images. (**F**) SDS-PAGE analysis of protein samples from a high-speed F-actin co-sedimentation experiment. F-actin, 3 μM; ADF7, 20 μM; ADF7^S128A, 20 μM; ADF7^S128D, 20 μM. The original pictures are available in S1 Raw Images. (**G**) Quantification of the amount of actin in the supernatant fractions in (**F**). Data are presented as mean ± SE, $n = 3$, ns, no significant difference, *$P < 0.05$ by Student's $t$ test. Numerical data underlying this panel are available in S4 Data. (**H**) Kinetic actin filament depolymerization assay. Actin filaments were depolymerized more rapidly by ADF7^S128D than by ADF7 and ADF7^S128A. Briefly, 5 μM pre-clarified ADF7, ADF7^S128A, or ADF7^S128D were incubated with 5 μM preassembled actin filaments (50% NBD-labeled) for 2 min. The mixtures were subsequently diluted 25-fold into buffer G and actin depolymerization was monitored by measuring the changes in fluorescence. Numerical data underlying this panel are available in S4 Data. (**I**) Images of actin filaments stained with Rhodamine-Phalloidin. The concentration of preassembled actin filaments is 2 μm. Bar = 10 μm. (**J**) Quantification of the average length of actin filaments shown in (**I**). The average length of actin filaments was plotted, and the data are presented as mean ± SE, $n = 3$, ns, no significant difference, **$P < 0.01$ (Student's $t$ test). Numerical data underlying this panel are available in S4 Data. ADF, actin-depolymerizing factor; CDPK, calcium-dependent protein kinase; LC-MS, liquid chromatography–mass spectrometry; WT, wild type.

To directly visualize the effect of phosphorylation of ADF7 at its Ser128 on the actin cyto-skeleton in pollen tubes, we directly visualized the actin cytoskeleton in pollen tubes after staining with Alexa-488 phalloidin. We found that actin filaments were overall brighter in *proADF7::gADF7^S128A; adf7 adf10* and *proADF7::gADF7^S128D; adf7 adf10* pollen tubes than in *adf10* and *proADF7::gADF7; adf7 adf10* pollen tubes (Fig 5A), and the increase in the amount of actin filaments is much more obvious within the apical and subapical regions of pollen tubes (Fig 5A and 5B). Accordingly, the average fluorescence intensity of phalloidin staining within the apical and subapical regions of *proADF7::gADF7^S128A; adf7 adf10* and *proADF7::gADF7^S128D; adf7 adf10* pollen tubes is significantly higher than in *adf10* and *proADF7::gADF7; adf7 adf10* pollen tubes (Fig 5C–5E), whereas no significant difference was detected in the shank region (Fig 5C and 5F). The actin cytoskeleton exhibits severe turnover defects in the apical and subapical regions but not in the shank region of pollen tubes harboring ADF7^S128A and ADF7^S128D, which suggests that phosphorylation of ADF7 at Ser128 has bio-logically meaningful consequence within apical and subapical regions that have high concen-tration of cytosolic Ca^2+. This, on the other hand, suggests that CDPK16-mediated regulation of ADF7 is biological significant in pollen tubes.

## CDPK16 mainly localizes to the plasma membrane in pollen tubes

To determine the intracellular localization of CDPK16 in pollen tubes, we generated a CDPK16-eGFP fusion construct with its expression under the control of the *CDPK16* native promoter. CDPK16-eGFP can rescue the LatB-resistant pollen germination phenotype of *cdpk16* mutants (S13 Fig), which suggests that CDPK16-eGFP is functional. We found that CDPK16 is mainly localized in an inner circle close to the border of ungerminated pollen grains (Fig 6A). During the onset of pollen germination, the CDPK16-eGFP signal is reduced at the germination aperture and is subsequently enriched in the region near the PM of the emerging pollen tube (Fig 6A and S9 Movie). We also examined the intracellular localization of CDPK16-eGFP in late-stage pollen tubes and found that it mainly localized to the PM, with the strongest signal in the subapical region (Fig 6B and S10 Movie). CDPK16-eGFP also forms small dots within the cytoplasm of pollen tubes (Fig 6B). We do not currently know what those structures are. The PM localization of CDPK16-eGFP was also confirmed by covisualization of the fluorescent lipophilic dye FM4-64. CDPK16-eGFP colocalized with FM4-64, and the eGFP signal was obviously stronger at the subapical region of the pollen tube (Fig 6C). We also found that CDPK16-eGFP is localized to the nucleus (Fig 6B and 6C). Our intracellular locali-zation data are actually consistent with a previous finding that CDPK localizes to membranes via myristoylation and palmitoylation [35].

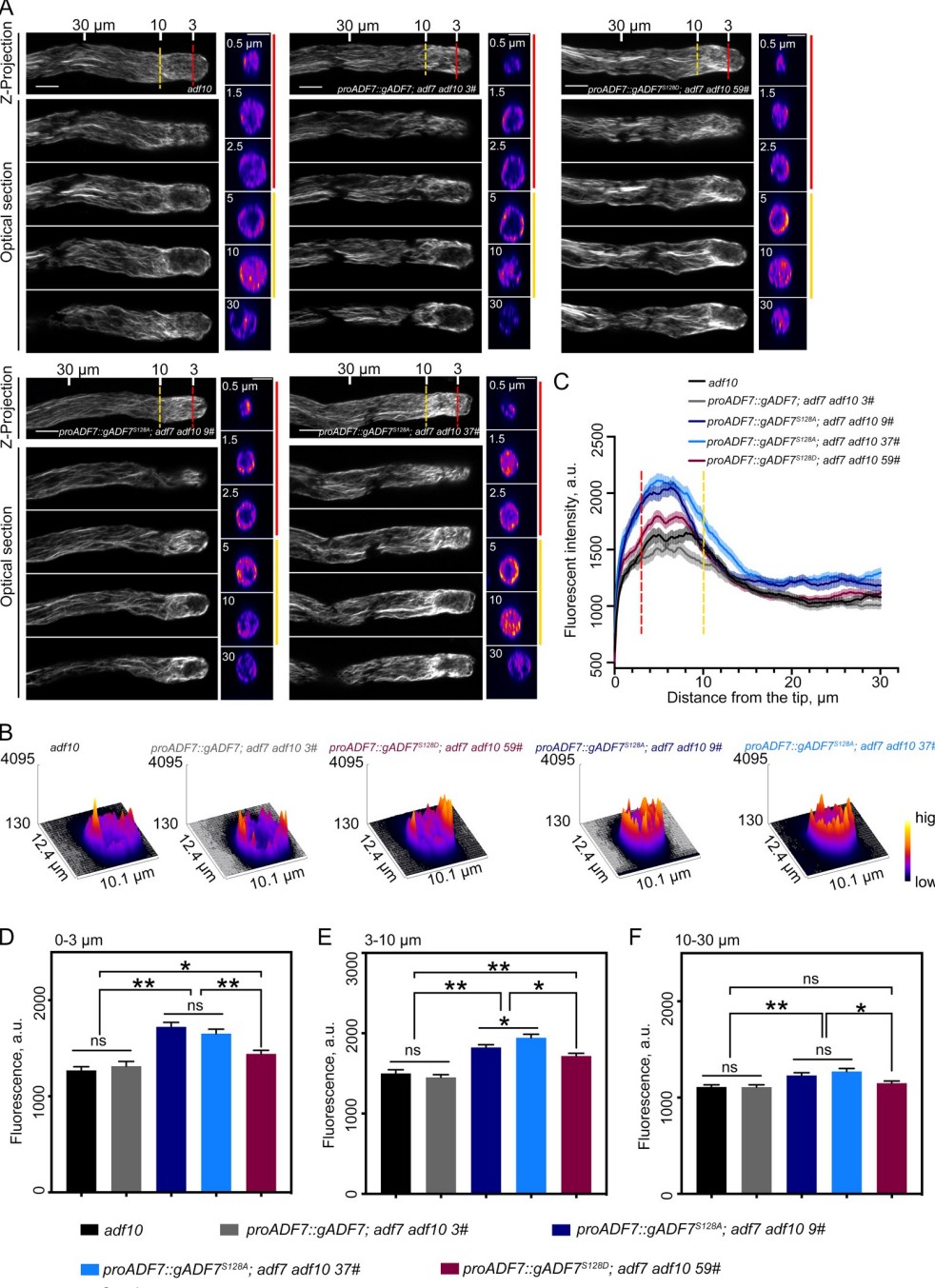

**Fig 5. ADF7$^{S128A}$ has less activity than ADF7 in restoring the actin turnover defects caused by loss of *ADF7* in pollen tubes.** (**A**) Images of actin filaments in pollen tubes. Pollen tubes derived from *adf10*, *proADF7::gADF7; adf7 adf10*, *proADF7::gADF7$^{S128A}$; adf7 adf10* and *proADF7::gADF7$^{S128D}$; adf7 adf10* were subjected to actin staining with Alexa-488 phalloidin. The projection images and the associated optical sections are presented. Several transverse sections derived from the same pollen tube are shown in the right panels; the distance of these sections from the pollen tube tip is indicated in the images. The dashed red lines indicate the base of the apical region (3 μm from tube tip), and the dashed yellow lines indicate the base of the subapical region (10 μm from tube tip). Bars = 5 μm. (**B**) The 3D distribution of fluorescence intensity of actin filaments within the apical region (0–3 μm from tube tip), which was generated by ImageJ software with a 3D interactive "Surface Plot" function. Warm and cold colors indicate higher and lower fluorescence, respectively. Numerical data underlying this panel are available in S5 Data. (**C**) Quantification of the fluorescence intensity of actin filaments stained with Alexa-488 phalloidin in pollen tubes. The fluorescence intensity (mean ± SEM) was plotted from the tip to the base along the growth axis of pollen tubes. The dashed red lines indicate the base of the apical region (3 μm from tube tip), and the dashed yellow lines indicate the base of the

subapical region (10 μm from tube tip). More than 40 pollen tubes were measured. Data are presented as mean ± SEM. Numerical data underlying this panel are available in S5 Data. (**D–F**) Quantification of the fluorescence intensity of Alexa-488 phalloidin within 3 different regions of pollen tubes. Data are presented as mean ± SEM, $n = 3$, ns, no significant difference, $*P < 0.05$, $**P < 0.01$ by Student's $t$ test. Numerical data underlying this panel are available in S5 Data. ADF, actin-depolymerizing factor.

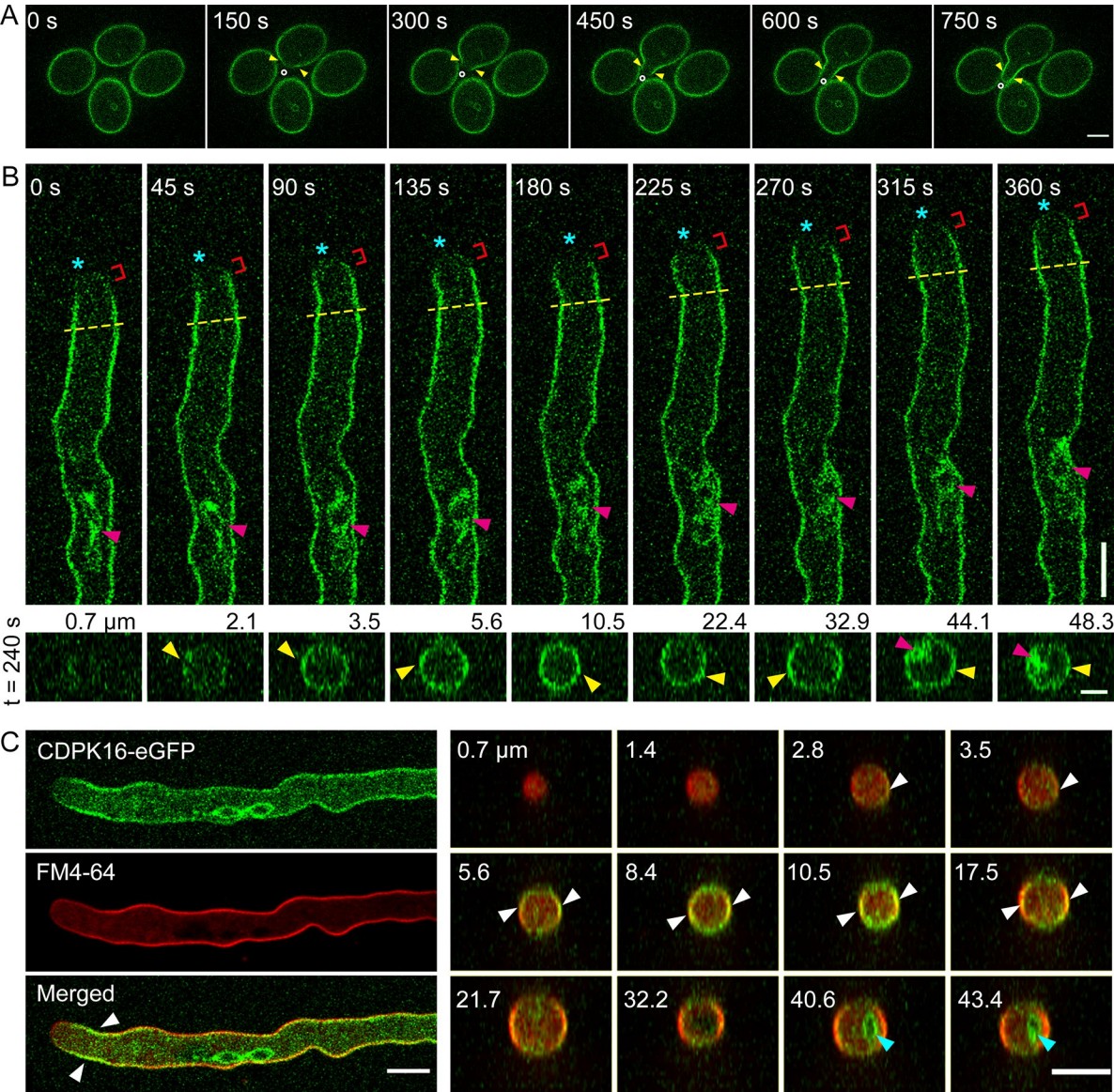

**Fig 6. Subcellular localization of CDPK16.** (**A**) Intracellular localization of CDPK16-eGFP in pollen grains during germination. Yellow triangles indicate the germination aperture, and white circles indicate the growth direction of the germinating pollen tube. Bar = 10 μm. (**B**) Intracellular localization of CDPK16-eGFP in pollen tubes. The upper panel is the time-lapse images showing the intracellular localization of CDPK16-eGFP in growing pollen tubes. Bar = 10 μm. The lower panel shows transverse sections at 240 s. Bar = 5 μm. Asterisks indicate the growth direction of the pollen tube, and red brackets indicate the apical region with less CDPK16-eGFP signals. The dashed yellow lines indicate the base of the subapical region. The red triangles indicate the membrane of the vegetative nucleus and the yellow triangles indicate the PM. The distance of transverse sections from the tip is indicated above the images. (**C**) Covisualization of CDPK16-eGFP with FM4-64 in the pollen tube. FM4-64 labels the PM but not internal membranes. The right panel shows transverse sections with their distance from the tip indicated in the images. White triangles indicate the PM and blue triangles indicate the membrane of the vegetative nucleus. Bars = 10 μm. CDPK, calcium-dependent protein kinase; PM, plasma membrane.

## Discussion

We here demonstrate that CDPK16 promotes actin turnover at least partly through the up-regulation of ADF7 activity in pollen. Our *in vitro* biochemical data show that CDPK16 phosphorylates Ser128 in ADF7 and up-regulates its actin-depolymerizing and severing activities. These findings differ from previous demonstrations or assumptions that kinase-mediated phosphorylation of plant ADFs occurs at Ser6 and down-regulates ADF activity [29–32,36,37]. Our study is the first to show that kinase-mediated phosphorylation up-regulates ADF/cofilin activity. Our results suggest that ADF7 is well suited to the regulation of actin dynamics within the apical region of pollen tubes, which harbor a tip-high $Ca^{2+}$-gradient.

ADF/cofilin is an essential regulator of actin dynamics, and therefore, there is great interest in the mechanisms that control its activity. It is well known that kinase-mediated phosphorylation is one of the major mechanisms for inactivating ADF/cofilin [38]. This inactivating phosphorylation occurs at Ser3 in animals and has been demonstrated both *in vitro* [39] and *in vivo* [40]. A similar mechanism has been demonstrated for plant ADFs, with the phosphorylation occurring at Ser6 [29]. The plant community takes it for granted that phosphorylation-mediated regulation of plant ADFs is achieved by adding a phosphate group to Ser6. Therefore, researchers normally manipulate Ser6 in order to abolish phosphorylation-mediated regulation of plant ADFs *in vivo* [9,37]. We found that Ser6 is also highly conserved among class II ADFs (S8 Fig), but we did not detect any ADF7 peptides containing phosphorylated Ser6 during our mass spectrometry analyses. This suggests that the frequency of Ser6 phosphorylation in ADF7 is comparatively low in pollen, even if this phospho-regulation mechanism does apply to ADF7. However, we repeatedly identified the ADF7 peptide containing phosphorylated Ser128 (Fig 4A), which suggests that the phosphorylation of ADF7 at Ser128 frequently occurs *in vivo*. In support of this notion, we found that treatment with phosphatase reduced the amount of phosphorylated ADF7 in total pollen extract probed with anti-phospho-ADF7 (Ser128) antibody (S9D Fig). In support of our observations, a recent report showed that CPK3-mediated phosphorylation of the C-terminus of ADF4 links actin remodeling to pattern-triggered immunity and effector-triggered immunity [41], but the biochemical mechanism underlying this regulation remains unknown. The biological significance of ADF7[S128] phosphorylation is supported by data showing that ADF7[S128A] retains roughly the same actin-depolymerizing activity as ADF7 *in vitro* (Fig 4F–4J), but has reduced activity in promoting actin turnover within the apical and subapical regions of pollen tubes (Figs 5 and S11). Surprisingly, we found that ADF7[S128D] also has reduced activity in promoting actin turnover in pollen (S11 Fig), suggesting that ADF7[S128D] cannot fully mimic the function of phosphorylated ADF7 with the phosphorylation occurring at its Ser128. In support of this speculation, we found that ADF7[S128D] only has slightly enhanced activity in depolymerizing and severing actin filaments compared to ADF7 *in vitro* (Fig 4), whereas incubation of ADF7 with CDPK16 dramatically enhanced the actin-depolymerizing activity of ADF7 (Fig 3), albeit only a part of ADF7 protein is supposed to be phosphorylated by CDPK16 under the same condition based on the western blot results probed with anti-phospho-ADF7(Ser128) (S9A Fig). ADF7[S128D] cannot fully represent phosphorylated ADF7 in depolymerizing actin filaments, which could be due to the possibility that Ser128 in ADF7 is not the only residue targeted by CDPK16. In the future, identification of other potential residue(s) in ADF7 that might be targeted by CDPK16 will help to clarify whether this possibility does exist. Up-regulation of the activity of class II ADFs by phosphorylating Ser128 might be a universal mechanism in plants, as Ser128 is highly conserved among class II ADFs (S8 Fig). Surprisingly, based on the results from 2D gels (Fig 2F and 2G), ADF7 seems to be preferentially subjected to phosphorylation in pollen when compared to ADF10. We do not know currently how this selective phosphorylation is achieved.

Nonetheless, we report a completely new mechanism for regulating the activity of ADF/cofilin in which kinase-mediated phosphorylation enhances the activity of ADF7 in *Arabidopsis*. Interestingly, we found that CKL2 also enhances the actin-depolymerizing activity of ADF7 (S14 Fig). We do not know why CKL2 has opposing effects on the actin-depolymerizing activity of ADF4 and ADF7, as CKL2 inhibits the actin-depolymerizing activity of ADF4 [30]. Nonetheless, this observation suggests that regulation of ADF7 by different kinases might allow actin dynamics to be integrated into different signal transduction pathways.

Since the discovery that ADF is phosphorylated and inactivated by CDPK(s) in plants [32], researchers in this field have been confused about how ADF(s) enhance actin dynamics at the tip of pollen tubes and root hairs, where the $Ca^{2+}$ concentration is high and CDPKs are supposed to be constitutively active. One of the explanations proposed by researchers is that dephosphorylated ADF is redistributed to the tip of root hairs via an unknown mechanism to promote actin turnover [29]. Our findings that CDPK16-mediated phosphorylation enhances the activity of ADF7 (Fig 3) suggest that ADF7 is well suited to enhancing actin turnover locally at pollen tube tips. In support of this notion, we found that *cdpk16* mutant pollen tubes exhibit more severe defects in actin turnover at pollen tube tips (Fig 1) and ADF7[S128A] has reduced activity in promoting actin turnover at pollen tube tips (Fig 5), although ADF7[S128A] has roughly the same actin-depolymerizing activity as ADF7 *in vitro* (Fig 4F–4J). We found that CDPK16 is concentrated on the membrane at the subapex but is comparatively less on the membrane at the apex (Fig 6), suggesting that the majority of CDPK16-mediated phosphorylation of ADF7 mainly occurs at the subapex. However, the cytoplasmic phosphorylated ADF7 should be able to rapidly diffuse to the apex to promote actin turnover, explaining why the rate of actin turnover was reduced at pollen tip including apical and subapical regions (Fig 1). Nonetheless, our findings completely change our view of how ADF contributes to the enhancement of actin dynamics within the apical region of pollen tubes, where there is a high $Ca^{2+}$ concentration that will in turn activate CDPK(s). Based on our *in vitro* and *in vivo* data, we propose that CDPK16 phosphorylates ADF7 and enhances its actin severing and depolymerizing activity to promote actin turnover within the apical region of pollen tubes where the free $Ca^{2+}$ concentration can reach the micromolar range [42,43]. As such, CDPK16 is involved in controlling the length and spatial distribution of apical actin filaments generated by membrane-anchored formins (Fig 7). Our findings thus suggest a scenario that unifies the high $Ca^{2+}$-CDPK activity with ADF activation to maintain the high dynamics of actin filaments at pollen tube tips.

As a universal second messenger, $Ca^{2+}$ is involved in numerous physiological cellular processes in eukaryotes. The cytosolic concentration of $Ca^{2+}$ ($[Ca^{2+}]_{cyt}$) undergoes rapid changes in response to various internal and external stimuli, which are normally accompanied by the rapid rearrangement of actin filaments [44]. The $Ca^{2+}$ levels are sensed by different $Ca^{2+}$-binding proteins, such as the conserved eukaryotic protein calmodulin (CaM) or proteins carrying a CaM-like domain. Among them, CDPKs represent the largest subfamily of $Ca^{2+}$ sensors in plants and have been implicated in different aspects of plant physiology [45–51]. Although several lines of evidence in the literature link CDPK signaling to the actin cytoskeleton [41,49,52], the underlying molecular mechanisms remain to be established. Our finding that CDPK16 promotes actin turnover by up-regulating the activity of ADF7 provides a mechanistic link between $Ca^{2+}$ signaling and rapid actin rearrangement in pollen tubes or in other plant cells in general. In this way, actin can be reorganized in response to environmental and developmental signals mediated by fluxes in $[Ca^{2+}]_{cyt}$. Our study thus enhances our understanding of the link between $Ca^{2+}$ signaling and actin dynamics in plants.

In summary, we report for the first time that kinase-mediated phosphorylation up-regulates the activity of ADF/cofilin. Our findings significantly enhance our understanding of the regulation of ADF/cofilin and suggest a scenario in which high $Ca^{2+}$/CDPK activity can be unified

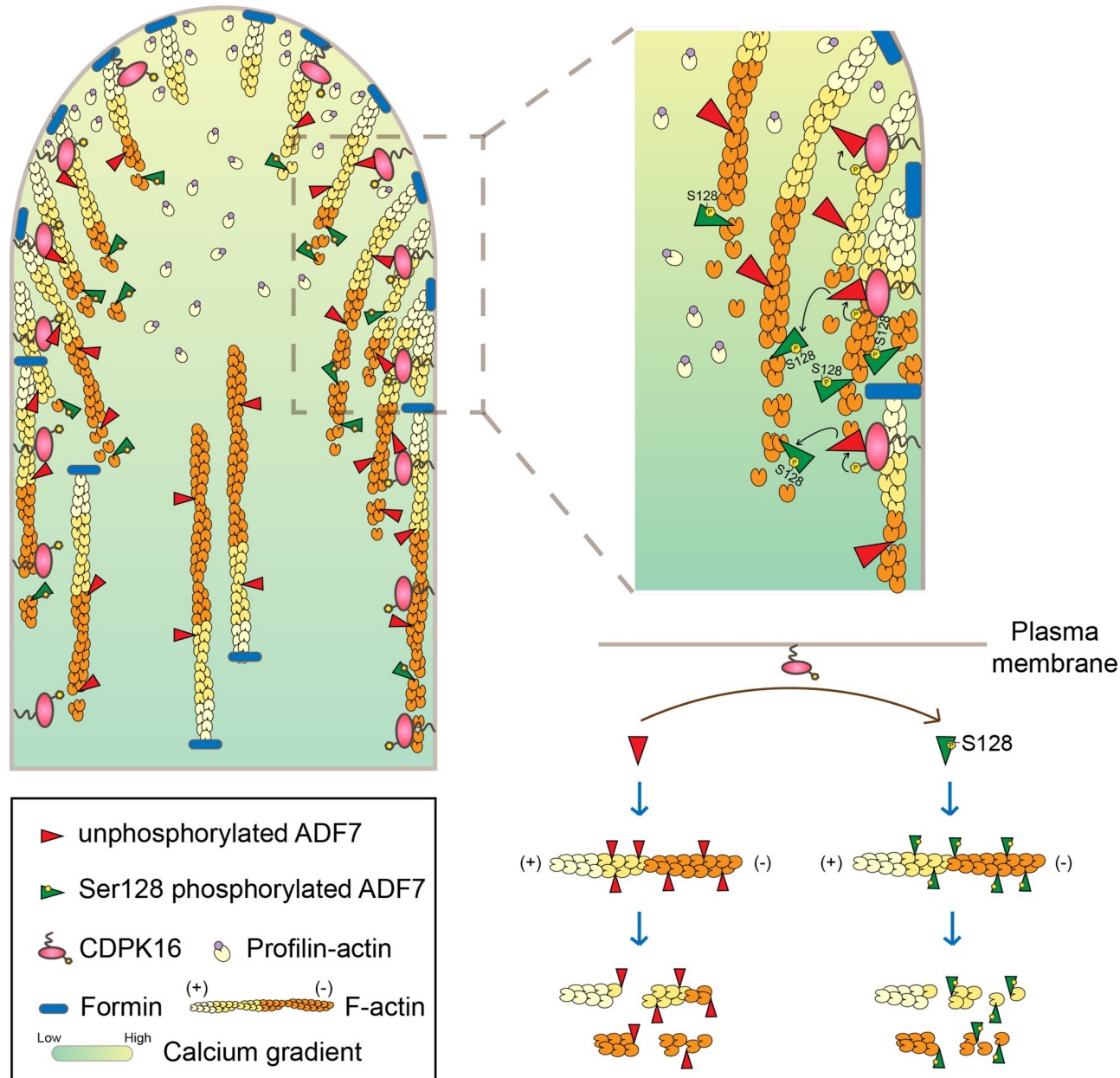

**Fig 7. Schematic model illustrating the role of CDPK16 in regulating the activity of ADF7 and actin dynamics within the growth domain of pollen tubes.** The left panel shows the organization of actin filaments within the apical and subapical regions of the pollen tube, where actin filaments are mainly polymerized from the PM by membrane-anchored formins utilizing profilin-actin complexes in the cytoplasm [60,74,75]. ADF has been implicated in the regulation of the turnover of apical actin filaments via severing and depolymerization [11]. This limits the length of filaments and shapes their organization to form the unique "apical actin structure" [6,7]. We found here that the PM-localized CDPK16 is also involved in promoting the turnover of those PM-originated actin filaments via phosphorylation of ADF7 at Ser128 to enhance its actin severing and depolymerizing activity. The boxed region in the left panel, where CDPK16 is comparatively concentrated, is enlarged in the upper right panel. The schematic diagram in the lower right panel shows that phosphorylation of Ser128 in ADF7 enhances its actin severing and depolymerizing activity. ADF, actin-depolymerizing factor; CDPK, calcium-dependent protein kinase; PM, plasma membrane.

with ADF activation to promote actin dynamics at pollen tube tips or root hair tips. Our results shed light on the molecular mechanisms underlying the regulation of actin dynamics during polarized growth of tip-growing cells and have general implications for understanding the dynamic interplay between Ca$^{2+}$ signaling and actin dynamics in plants.

## Materials and methods

### Plant materials and growth conditions

The T-DNA insertion mutant of *CDPK16*, Salk_052257, was obtained from the Arabidopsis Biological Resource Center (ABRC) and was designated as *cdpk16-1*. Another mutant of *CDPK16* designated as *cdpk16-2* was generated by the CRISPR/Cas9 system (see below). Information on the *adf7*, *adf10* and *adf7 adf10* mutants has been described previously [10,11]. *ADF7* and *CDPK16* overexpressors were generated as described below. The *adf10 cdpk16-1* double mutants were generated by crossing *adf10* with *cdpk16-1*. To examine the effect of *ADF7* gain-of-function on *cdpk16*, *cdpk16-1* was crossed with *ADF7* overexpressors. To observe actin dynamics in *cdpk16* pollen tubes, the probe Lifeact-eGFP was introduced into *cdpk16-1* via crossing with WT *Arabidopsis* plants expressing *Lat52::Lifeact-eGFP* [5]. *ADF7* genomic DNA was amplified using the primer pair PG3/PG4 (S1 Table). This PCR product, with 8His fused to the N-terminus of ADF7, was moved into pCAMBIA1301 to generate pCAMBIA1301-*pADF7-8His-gADF7*. The pCAMBIA1301-*pADF7-8His-gADF7* plasmid was subsequently transformed into *adf7* using the *Agrobacterium*-mediated floral dip method [53]. The transgenic plants were designated as *proADF7::8His-gADF7; adf7* and were used at T3. *Arabidopsis* Columbia-0 ecotype (Col-0) was used as WT. *Arabidopsis* plants were grown in media or soil at 22°C under a 16-h-light/8-h-dark photoperiod.

### Generation of the *CDPK16* knockout mutant using the CRISPR/Cas9 system in *Arabidopsis*

The egg cell-specific promoter-controlled (EPC) CRISPR/Cas9 system was employed to generate *CDPK16* loss-of-function mutants as described previously [33]. Briefly, the plasmid pHEE401-*CDPK16* was constructed using primer pairs PG3/PG4 and PG5/PG6 (S1 Table) and transformed into WT *Arabidopsis* plants to generate *CDPK16* loss-of-function mutants. U6_26p-CDPK16_sgRNA1-gRNA_Sc-U6_26t-U6_29p-CDPK16_sgRNA2-gRNA_Sc-U6_26t was included in the plasmid pHEE401-*CDPK16*. Total DNA was extracted from the leaves of T1 transgenic plants and the target sequence was then amplified by primer pair PG7/PG8 (S1 Table). Sequencing was performed to verify the mutation. The *CDPK16* loss-of-function mutant generated by CRISPR/Cas9 was designated as *cdpk16-2*.

### Creation of *ADF7* and *CDPK16* overexpressors

To generate *ADF7* overexpression transgenic plants, the coding region sequence (CDS) of *ADF7* was amplified with primer pair O1/O2 (S1 Table) using pET28a-ADF7 [10] as the temperate. The *ADF7* CDS was moved into pCAMBIA1301-*Lat52* to generate pCAMBIA1301-*Lat52-ADF7*. To generate *CDPK16* overexpressors, the *CDPK16* CDS was amplified with primer pair O3/O4 (S1 Table) and subsequently moved into pCAMBIA1301-*Lat52* to generate pCAMBIA1301-*Lat52-CDPK16*. The plasmid pCAMBIA1301-*Lat52-ADF7* or pCAMBIA1301-*Lat52-CDPK16* was transformed into *Arabidopsis* using the *Agrobacterium*-mediated floral dip method [53]. To generate *CDPK16* overexpressors in the *proADF7::gADF7; adf7 adf10* and *proADF7::gADF7^{S128A}; adf7 adf10* backgrounds, the fragment of *CDPK16* CDS fused with the Lat52 promoter at its N-terminus was amplified from pCAMBIA1301-*Lat52-CDPK16* plasmid with primer pair O5/O6 (S1 Table) and subsequently moved into pK7FWG2 to generate pK7FWG2-*Lat52-CDPK16*. The plasmid was then transformed into *proADF7::gADF7; adf7 adf10* and *proADF7::gADF7^{S128A}; adf7 adf10* transgenic plants. T3 homozygous transgenic plants were used for subsequent analysis.

## Complementation of *cdpk16* mutants and visualization of the subcellular localization of CDPK16 in pollen tubes

To complement *cdpk16* mutants, the genomic DNA sequence of *CDPK16* containing a 3.4-kb predicted promoter sequence was amplified with primer pair PG1/PG2 (S1 Table). The DNA fragment was cloned into pEasy-Blunt to generate pEasy-Blunt-*pgCDPK16* and was subsequently moved into pCAMBIA1301 digested with *Pst*I/*Kpn*I to generate the complementary plasmid pCAMBIA1301-*pgCDPK16*. Subsequently, eGFP was moved into pCAMBIA1301-*pgCDPK16* digested with *Sac*I/*Eco*RI to generate the plasmid pCAMBIA1301-*pgCDPK16-eGFP*. The plasmid pCAMBIA1301-*pgCDPK16-eGFP* was transformed into *cdpk16-1* and *cdpk16-2* to generate transgenic plants designated as *proCDPK16::gCDPK16-eGFP; cdpk16-1* and *proCDPK16::gCDPK16-eGFP; cdpk16-2*, respectively. To determine the subcellular localization of CDPK16, *proCDPK16::gCDPK16-eGFP; cdpk16-2* pollen tubes were observed under an Olympus FluoView FV1200 confocal microscope equipped with a 100× oil immersion objective (1.4 numerical aperture). Samples were excited under a 488-nm argon laser with the emission wavelengths set at 505 to 545 nm. The z-series images were collected with the z-step size set at 0.5 μm.

## Determination of the effect of amino acid substitution at serine-128 of ADF7 in pollen tubes

To assay the effect of single amino acid substitution at Serine-128 of ADF7, the plasmid pEasy-Blunt-pgADF7 was used as the template to perform PCR amplification with the primer pairs M1/M2 and M3/M4 (S1 Table) to generate pEasy-Blunt-*pgADF7*$^{S128A}$ and pEasy-Blunt-*pgADF7*$^{S128D}$, respectively. The WT and mutant inserts were subsequently moved into pFGC5941 to generate the plasmids pFGC5941-*pgADF7*, pFGC5941-*pgADF7*$^{S128A}$, and pFGC5941-*pgADF7*$^{S128D}$. The plasmids pFGC5941-*pgADF7* and pFGC5941-*pgADF7*$^{S128A}$ and pFGC5941-*pgADF7*$^{S128D}$ were transformed into *adf7-/-adf10+/-* using the *Agrobacterium*-mediated floral dip method [53]. After self-segregation, *adf7 adf10* plants containing the homozygous pFGC5941-*pgADF7*, pFGC5941-*pgADF7*$^{S128A}$, and pFGC5941-*pgADF7*$^{S128D}$ were obtained, and they were designated as *proADF7::gADF7; adf7 adf10* and *proADF7::gADF7*$^{S128A}$; *adf7 adf10*, respectively.

## RNA extraction and qRT-PCR analysis

Total RNA was extracted from *Arabidopsis* mature pollen with a Total RNA Extraction Kit (Promega, LS1040). Subsequently, total RNA together with Primers Oligo (dT)$_{18}$ and M-MLV reverse transcriptase (Promega, M1075) were used for reverse transcription to synthesize cDNA. Quantitative real-time PCR (qRT-PCR) was conducted using 2× RealStar Green Power Mixture with ROX II (GenStar, A314-10). To determine the transcript levels of *CDPK16* in WT, *CDPK16* overexpressors and the *cdpk16-1* mutant, a partial coding region sequence (CDS) of *CDPK16* was amplified with primer pair R3/R4 (S1 Table) by qRT-PCR. *eIF4A* was amplified as an internal control with primer pair R1/R2 (S1 Table). To confirm the transcript levels of *ADF7* in *proADF7::gADF7; adf7 adf10*, *proADF7::gADF7*$^{S128A}$; *adf7 adf10* and *gADF7*$^{S128D}$;*adf7 adf10* plants, a partial CDS of *ADF7* was amplified with primer pair R5/R6 (S1 Table) by qRT-PCR. The relative amounts of *ADF7* and *CDPK16* transcripts were quantified by the $2^{-\Delta\Delta Ct}$ method [54].

## Western blot analysis and quantification of the relative amount of ADF7 and ADF7$^{S128A}$ protein in pollen

To determine the amount of ADF7 and its variant ADF7$^{S128A}$ or ADF7$^{S128D}$ in pollen, total proteins were isolated from *Arabidopsis* pollen grains derived from *proADF7::gADF7; adf7 adf10, proADF7::gADF7$^{S128A}$; adf7 adf10* and *proADF7::gADF7$^{S128D}$; adf7 adf10* plants according to previously published methods [55,56]. ADF7 and its variant ADF7$^{S128A}$ were detected in pollen by probing with anti-ADF antibody (ADF7 polyclonal antibody) as described previously [24], and the relative amount of ADF7 was normalized to the amount of UGPase probed with anti-UGPase antibody (Agrisera, AS05086).

## Two-dimensional electrophoresis assay

Two-dimensional (2D) electrophoresis assay was performed roughly according to a previously published method [30]. Briefly, total proteins were isolated from mature pollen grains derived from WT, *adf7* (Salk_024576) [10], *adf10* [11], *proADF7::gADF7$^{S128A}$; adf7, proADF7:: gADF7$^{S128D}$; adf7 cdpk16* and *CDPK16* overexpressor plants, as described previously [55,56]. Briefly, mature pollen grains were finely ground in liquid nitrogen and the powder was mixed with protein extraction buffer [40 mM Tris-HCl (pH 7.5), 60 mM DTT, 2% SDS, cOmplete, EDTA-free Protease Inhibitor Cocktail (Sigma-Roche, 4693159001), and 1% Phosphatase Inhibitor Cocktail 2 (Sigma-Aldrich, P5726)]. For phosphatase treatment, total proteins were extracted in protein extraction buffer without cOmplete, EDTA-free Protease Inhibitor Cocktail 2. The proteins were then incubated with Lambda Protein Phosphatase (λpp, New England Biolabs, P0753S) at 30°C for 30 min. Then, the mixtures were incubated at 95 to 100°C for 5 min and centrifuged at 12,000 rpm for 10 min. The total pollen proteins were in the supernatant. The proteins were further purified using the 2-D Clean-Up kit and protein pellets were obtained (GE Healthcare, 80-6484-51). The pellets were re-dissolved with 2-DE buffer (8 M Urea, 2.5 M Thiourea, 4% CHAPS, 65 mM DTT, and 1% IPG buffer) and centrifuged at 13,000 g for 10 min. Then, 20 μl supernatant was separated and mixed with 115 μl Hydration buffer (8 M Urea, 2% CHAPS, 65 mM DTT, 1% IPG buffer, and 0.001% Bromophenol blue). Subsequently, iso-electric focusing and 15% SDS-PAGE were performed in a Protein I12 system chamber (Bio-Rad). After electrophoresis, immunoblot analysis was performed with anti-ADF7 antibody [24].

## Production of anti-phospho-ADF7(Ser128) antibody and detection of phosphorylated ADF7 in pollen

To generate the poly-clonal antibody that specifically recognizes the ADF7 phosphorylated at Ser128, a phosphorylated peptide (ELDGIQVELQATDPSEM(P)SFDIIK) was synthesized and used to immunize rabbit to generate the antibody designated as anti-phospho-ADF7(Ser128). The specificity of the antibody was examined by performing western blot analysis after incubation of ADF7 with CDPK16 in the presence or absence of calcium. To detect the phosphorylated ADF7 in pollen, total proteins were isolated from pollen derived from *proADF7::8His-gADF7; adf7* plants. To perform phosphatase treatment, Lambda Protein Phosphatase (λpp, New England Biolabs, P0753S) was added into the extraction buffer. The total protein extract was subsequently incubated with Ni-NTA agarose that was washed extensively with protein extraction buffer after centrifugation. The protein bound to the Ni-NTA agarose was separated by 15% SDS-PAGE and subjected to western blot analysis probed with Anti-phospho-ADF7 (Ser128). The relative amount of loaded protein was determined by performing western blot analysis using anti-ADF7 antibody [24]. The total pollen extract without λpp treatment was used as the control.

### *In vitro Arabidopsis* pollen germination and pollen tube growth

*In vitro Arabidopsis* pollen germination and pollen tube growth assays were performed as described previously [57]. Briefly, pollen grains were cultured under moist conditions at 28˚C on pollen germination medium [PGM: 1 mM $MgSO_4$, 1 mM $CaCl_2$, 1 mM $Ca(NO_3)_2$, 0.01% (w/v) $H_3BO_3$, and 18% (w/v) Suc, and 0.8% (w/v) agarose (pH 6.9 to 7.0)]. To determine the effect of Latrunculin B (LatB, Sigma-Aldrich, L5288) treatment on pollen germination, pollen grains were cultured on GM in the absence or presence of 1.5 nM or 3 nM LatB for 3 h and observed under an Olympus CX21 microscope equipped with a 10× objective. More than 500 pollen grains were counted in each experiment and the experiments were conducted at least 3 times. To determine the effect of LatB on the growth of pollen tubes, liquid PGM in the absence or presence of 3 nM LatB was added onto the surface of solid GM. To determine the velocity of pollen tube growth, the length of pollen tubes was measured at 2 different time points and the extension was divided by the time interval to yield the average pollen tube growth rate. More than 30 pollen tubes were measured in each experiment and the experiments were repeated at least 3 times.

### Screening for *Arabidopsis* T-DNA insertion mutants with a LatB-resistant pollen germination phenotype

Confirmed *Arabidopsis* homozygous T-DNA insertion lines were obtained from the Nottingham Arabidopsis Stock Centre (NASC). To identify genes involved in the regulation of actin turnover in pollen, the T-DNA insertions lines were screened for a LatB-resistant pollen germination phenotype. Lines were selected in which the T-DNA insertion was located in genes that showed a reasonable level of expression in pollen, based on data from www.genevestigator.com. To identify the LatB-resistant pollen germination phenotype of the T-DNA insertion lines, 1.5 nM LatB was included in solid PGM. The detailed procedure for *Arabidopsis* pollen germination is described above.

### F-actin staining in fixed *Arabidopsis* pollen grains and pollen tubes

*Arabidopsis* pollen grains and pollen tubes were subjected to actin staining with Alexa-488 phalloidin (Thermo Fisher Scientific, A12379) as previously reported [58,59]. To determine the effect of LatB on the actin cytoskeleton, pollen grains or pollen tubes were treated with 150 nM LatB in liquid GM for 30 min before fixation with 300 µM 3-maleimidobenzoic acid *N*-hydroxysuccinimide ester crystalline (Sigma-Aldrich, M2786) and subsequent staining with Alexa-488 phalloidin. The samples were visualized under an Olympus FluoView FV1200 confocal microscope equipped with a 100× oil immersion objective (1.4 numerical aperture) and were excited under a 488-nm argon laser with the emission wavelength set at 505 to 545 nm. The z-series images were collected with the step size set at 0.5 µm. The amount of F-actin in pollen grains and pollen tubes was quantified by measuring the intensity of the fluorescent pixels with ImageJ software (http://rsbweb.nih.gov/ij/; version 1.48g) as reported previously [10].

### Visualization and quantification of the dynamics of actin filaments in pollen tubes

To observe the dynamics of actin filaments in pollen tubes, Lifeact-eGFP was introduced into pollen tubes as described previously [5]. Briefly, pollen tubes harboring Lifeact-eGFP were observed under the Olympus IX83 microscope equipped with a 60× oil immersion objective (1.35 numerical aperture). Time-lapse z-series images were acquired with an Andor Revolution XDh spinning disk confocal system using MetaMorph software (Molecular Devices) at

time intervals of 3 s and the z-step size set at 0.7 μm. To quantify the growth rate of pollen tubes and the fluorescence intensity of apical actin filaments simultaneously, a kymograph was produced using ImageJ software as previously reported [60]. To quantify the dynamics of individual actin filaments in pollen tubes, the parameters associated with them (actin filament elongation rate, actin filament depolymerization rate, maximum filament length, maximum filament lifetime, and severing frequency) were measured as described previously using ImageJ software [5,10].

## FM dye staining of living pollen tubes

Pollen tubes were stained with the Lipophilic Dye FM4-64 (Thermo Fisher Scientific, T13320). Staining of pollen tubes was achieved by adding FM4-64 dye (2.5 μM in liquid pollen germination medium) onto the surface of solid pollen germination medium. After incubation for 5 min, z-series images were collected by Olympus IX83 spinning disc confocal microscopy with the z-step size set at 0.7 μm. FM4-64 dye was excited with an argon laser at 561 nm, and the emission wavelength was set in a range of 600 to 650 nm.

## Luciferase complementation imaging (LCI)

To determine the interaction between CDPK16 and ADF7, the LCI assay was performed as described previously [61,62]. The coding region sequences of *CDPK16* and *ADF7* were fused with the N-terminus of LUC (nLUC) and the C-terminus of Luc (cLUC), respectively. The plasmids pCAMBIA1300-*CDPK16-nLUC* and pCAMBIA1300-*cLUC-ADF7* were transformed into *Agrobacterium tumefaciens* strain GV3101. Bacterial cells were centrifuged and re-suspended in infiltration buffer (10 mM MES-KOH (pH 5.7), 10 mM MgCl$_2$, 150 μM acetosyringone). After adjusting the OD600 of the bacterial suspensions to 0.6, the bacterial cells were subsequently co-infiltrated into *Nicotiana. benthamiana* leaves. After incubation at room temperature for 48 to 60 h, the luciferase (LUC) activity was measured with a cooled CCD imaging apparatus (Andor iXon, Andor Technology, Belfast, United Kingdom). The empty vector was used as a control.

## Protein production

The coding region sequence of *CDPK16* was amplified with primer pair P3/P4 (S1 Table) and moved into pET28a to generate pET28a-*CDPK16*. To generate ADF7$^{S128A}$, ADF7$^{S128D}$, ADF10$^{S128A}$, and ADF10$^{S128D}$ proteins, we initially generated plasmids pGEX-KG-*ADF7$^{S128A}$*, pGEX-KG-ADF7$^{S128D}$, pGEX-KG-ADF10$^{S128A}$, and pGEX-KG-ADF10$^{S128D}$ using pGEX-KG-*ADF7* and pGEX-KG-ADF10 plasmids [10,11] as the temperate with the primer pairs M1/M2, M3/M4, M5/M6 and M7/M8 (S1 Table), respectively. The plasmids pGEX-KG-ADF10, pGEX-KG-ADF10$^{S128A}$, and pGEX-KG-ADF10$^{S128D}$ were directly transformed into the *Escherichia coli* BL21 (DE3) strain. The sequences of *ADF7*, *ADF7$^{S128A}$*, and *ADF7$^{S128D}$* were then amplified with primer pair P1/P2 (S1 Table) using pGEX-KG-*ADF7*, pGEX-KG-*ADF7$^{S128A}$*, and pGEX-KG-*ADF7$^{S128D}$* as the templates and were moved into pET28a to generate pET28a-*ADF7*, pET28a-*ADF7$^{S128A}$*, and pET28a-*ADF7$^{S128D}$*, respectively. The resulting plasmids were transformed into the *E. coli* BL21 (DE3) strain. Generation of pET28a-ADF4 and pET28a-CKL2 plasmids was performed as described previously [30]. The recombinant proteins with His-tag or GST-tag were purified with Ni-NTA Resin (Merck Millipore, 70666–4) or Glutathione Sepharose 4B (GE Healthcare, 17-0756-05) according to the manufacturer's instructions. Actin was purified from rabbit skeletal muscle as described previously [63,64] and labeling of actin with Oregon Green 488 Iodoacetamide (Thermo Fisher Scientific,

O6010) or 7-chloro-4-nitrobenzo-2-oxa-1,3-diazole (NBD, Invitrogen, C20260) was performed as described previously [65,66].

## Kinase activity assay

*In vitro* kinase activity assays were performed to determine whether ADF is phosphorylated by CDPK16. Recombinant proteins CDPK16-6×His, ADF7-6×His, and ADF7$^{S128D}$-6×His were purified with Ni-NTA agarose. CDPK16-6×His (1 μg) was incubated with 1 μg ADF7-6×His or ADF7$^{S128D}$-6×His in a kinase reaction buffer (20 mM Tris-HCl (pH 8.0), 5 mM MgCl$_2$, 10 μM ATP (Jena Bioscience, NU-1103L), 1 mM DTT, and 2 μCi [γ-$^{32}$P] ATP) at 30°C for 30 min. The same volume of 2 × SDS loading buffer was added to terminate the phosphorylation reaction. The samples were analyzed by 15% SDS-PAGE and stained with Coomassie Brilliant blue R 250 (Sigma-Aldrich, C.I. 42660). The SDS-PAGE gel was exposed to a phosphor screen and the phosphorylation signals were detected the next day using a Typhoon 9410 phosphor imager (Amersham Biosciences). The phosphorylation signals were determined by densitometry using ImageJ software.

## Mass spectrometry

To identify the CDPK16-interacting proteins in pollen, Ni-NTA agarose bound with CDPK16-6×His was incubated with the total protein extract isolated from WT pollen. After 3 washes with protein extraction buffer, the proteins bound to the beads were separated by 15% SDS-PAGE and stained with Coomassie Brilliant blue R 250. To perform the mass spectrometry analysis, the gel bands containing the proteins of interest were first cut into pieces, and the remaining steps were performed according to the previously reported methods [67]. Briefly, mass spectrometry samples were prepared by decolorization, drying, reduction, alkylation, drying, enzymatic hydrolysis, termination and separation, extraction, reconstitution, etc. Finally, the samples were dissolved in 20 μl 0.1% formic acid. Mass spectrometry detection was performed using the Orbitrap Fusion Tribrid mass spectrometer (Thermo Fisher Scientific). To identify the phosphorylated site(s) of ADF7 *in vivo*, 8His-ADF7 was isolated from pollen grains derived from the transgenic line *proADF7::8His-gADF7; adf7* by Ni-NTA agarose. To identify the CDPK16-phosphorylated-site(s) of ADF7 *in vitro*, 20 μM ADF7 was incubated with 5 μM recombinant CDPK16-6×His in the kinase reaction buffer I (20 mM Tris-HCl (pH 8.0), 0.5 mM CaCl$_2$, 1 mM MgCl$_2$, 0.5 mM ATP, and 1 mM DTT) at 30°C for 30 min. After separation by 15% SDS-PAGE, the 8His-ADF7 band was cut out of the gel and analyzed by liquid chromatography–mass spectrometry/mass spectrometry (LC-MS/MS), which was performed at the Center of Biomedical Analysis, Tsinghua University.

## High-speed F-actin co-sedimentation assay

The high-speed F-actin co-sedimentation assay was adapted from previously published methods [68]. All proteins were pre-clarified at 200,000 g for 30 min at 4°C. Briefly, preassembled actin filaments (3 μM) were incubated with 20 μM ADF7, ADF7$^{S128A}$, or ADF7$^{S128D}$. After incubation for 1 h at 30°C, the reaction mixtures were subjected to centrifugation at 100,000 g for 30 min at 4°C. To determine the effect of CDPK16 and CKL2 on the actin-depolymerizing activity of ADF7, ADF7$^{S128A}$, and ADF4, varying concentrations of CDPK16 or CKL2 were incubated with their corresponding substrates ADF7 (20 μM) in kinase reaction buffer I (20 mM Tris-HCl (pH 8.0), 0.5 mM CaCl$_2$, 1 mM MgCl$_2$, 0.5 mM ATP, and 1 mM DTT) or kinase reaction buffer II (20 mM Tris-HCl (pH 8.0), 0.5 mM EGTA, 1 mM MgCl$_2$, 0.5 mM ATP, and 1 mM DTT) for 30 min at 30°C before incubation with 3 μM preassembled actin filaments at 30°C for 1 h. The reaction mixtures were subjected to centrifugation at 100,000 g for 30 min at

4°C. The protein samples in the supernatant and pellet fractions were separated by SDS-PAGE and revealed by staining with Coomassie Brilliant blue R 250. The amount of actin in the supernatant was quantified by densitometry using ImageJ software.

## Direct visualization of individual actin filaments by fluorescence light microscopy

The effect of ADF7 and its variants on shortening actin filaments was determined by fluorescence light microscopy as described previously [25]. Briefly, pre-polymerized actin filaments (2 μM) were incubated with 2.5 μM ADF7, 2.5 μM ADF7$^{S128A}$, or 2.5 μM ADF7$^{S128D}$ for 3 min at room temperature, and 5 μM Rhodamine-Phalloidin (Thermo Fisher Scientific, R415) was subsequently added to terminate the reaction and label actin filaments. Actin filaments were diluted to a final concentration of 100 nM with buffer F (15 μg/ml glucose, 0.5% methylcellulose, 100 μg/ml glucose oxidase, 20 μg/ml catalase, 50 mM KCl, 1 mM MgCl$_2$, 10 mM imidazole (pH 7.0), and 100 mM DTT) and were observed under an Olympus BX53 microscope equipped with a 1.42 NA ×60 oil immersion lens. The images were collected with an Olympus DP80 camera controlled by Cell Sens Standard 1.12 software. To determine the effect of CDPK16 on the activity of ADF7, various concentrations of CDPK16 were first incubated with ADF7 in kinase reaction buffer I at 30°C for 30 min. The remaining steps were the same as for ADF7 alone.

## Dilution-mediated actin depolymerization assay

To determine the ability of ADF7 and its variants on depolymerizing actin filaments, a dilution-mediated actin depolymerization assay was performed as described previously [69]. Briefly, 5 μM pre-centrifuged proteins (ADF7, ADF7$^{S128A}$, or ADF7$^{S128D}$) were incubated with 5 μM preassembled actin filaments (50% NBD-labeled) for 2 min at room temperature. The mixtures were subsequently diluted 25-fold into buffer G (5 mM Tris-HCl (pH 8.0), 0.2 mM CaCl$_2$, 0.02% NaN$_3$, 0.2 mM ATP, and 0.2 mM DTT). Actin depolymerization was traced by monitoring the changes in NBD fluorescence by the QuantaMaster Luminescence QM 3 PH Fluorometer (Photon Technology International) with the excitation and emission wavelengths set at 475 nm and at 530 nm, respectively.

## Direct visualization of actin filament dynamics by total internal reflection fluorescence microscopy (TIRFM) *in vitro*

The dynamics of individual actin filaments were visualized by TIRFM, which was roughly adapted from the previously published method [70]. The proteins used in this assay were pre-clarified at 200,000 g for 30 min at 4°C. The flow cell was firstly incubated with 100 nM N-ethylmaleimide-myosin for 3 min in the dark, followed by washing with 1% BSA for 2 min. The flow cell was then washed with 1×TIRFM buffer (10 mM imidazole (pH 7.0), 50 mM KCl, 1 mM MgCl$_2$, 1 mM EGTA, 50 mM DTT, 0.2 mM ATP, 50 μM CaCl$_2$, 15 mM glucose, 20 μg/ml catalase, 100 μg/ml glucose oxidase, and 0.5% methylcellulose). Actin filaments at 150 nM [50% Oregon Green 488-labeled] were injected into the chamber and incubated for 5 min at room temperature in the dark. The unbound actin filaments were removed by washing with 1×TIRFM buffer. ADF7, ADF7$^{S128A}$, ADF7$^{S128D}$, or ADF7 after incubation with CDPK16 in phosphorylation reaction buffer for 30 min at room temperature was injected into the chamber. Actin filaments were observed immediately by TIRF illumination with an Olympus IX-71 microscope equipped with a 100× oil objective (1.45 numerical aperture). The time-lapse images were collected for 5 min with a Hamamatsu ORCA-EM-CCD camera (model C9100-

12) driven by Micro-Manager software (www.micro-manager.org; v1.4; MMStudio) at time intervals of 3 s. More than 25 actin filaments (>10 μm) were chosen for quantification of the severing frequency (breaks/μm/s) of actin filaments with ImageJ software according to the previously published method [34].

## Sequence alignment

The sequences of ADF proteins from different plant species were obtained from NCBI referring to the previous phylogenetic analyses [71–73]. The sequence alignment was performed using the software MAGA7 (https://www.megasoftware.net) and the program ClustalW, and the resulting data were exported in the FASTA format. The FASTA format file was submitted to the online server ESPript3 (http://espript.ibcp.fr/ESPript/cgi-bin/ESPript.cgi) to predict the secondary structures using AtADF1 as the template and calculate the sequence similarities based on the %Equivalent method with the global score set to 0.7. The seqlogo diagram, which reflects the amino acid prevalence at each position in the peptide sequence encompassing Ser128, was generated by using the online software program Weblogo 3 (http://weblogo. threeplusone.com/create.cgi).

## Supporting information

**S1 Fig. Loss of function of *CDPK16* renders pollen germination and pollen tube growth resistant to LatB treatment.** (**A**) Gene structure of *CDPK16*. *CDPK16* contains 12 exons (black boxes) and 11 introns (black lines). The T-DNA insertion allele Salk_052257 was designated as *cdpk16-1*. The red triangle indicates the T-DNA insertion site. (**B**) Determination of the transcript level of *CDPK16* in WT and *cdpk16-1* by real-time quantitative RT-PCR. *eIF4A* was used as the internal control. Numerical data underlying this panel are available in S6 Data. (**C**) Creation of a *CDPK16* knockout allele by the CRISPR/Cas9 approach. The mutant allele was designated as *cdpk16-2*. Deletion of 2 bases, T and C, in the first exon of the *CDPK16* gene leads to early termination of protein translation (indicated by the asterisk). (**D**) Images of pollen grains and pollen tubes. Pollen derived from WT and *cdpk16* mutants were germinated on GM in the presence or absence of 1.5 nM LatB. Bar = 25 μm. (**E**) Plot of pollen germination rate in the presence of LatB. Data are presented as mean ± SE, **$P < 0.01$ by Student's *t* test. Numerical data underlying this panel are available in S6 Data. (**F**) Images of pollen tube growth at 2 time points. Single pollen tubes from WT and *cdpk16* mutants in the presence of 3 nM LatB were selected for measurement. Bar = 20 μm. (**G**) Quantification of pollen tube growth rate from (F) in the presence of LatB. Data are presented as mean ± SE, **$P < 0.01$ by Student's *t* test. Numerical data underlying this panel are available in S6 Data.
(TIF)

**S2 Fig. Loss of function of *CDPK16* renders the actin cytoskeleton resistant to LatB treatment in pollen grains.** (**A**) Micrographs of the actin cytoskeleton in pollen grains. Pollen grains of WT and *cdpk16* mutants were subjected to staining with Alexa-488 phalloidin in the presence or absence of 150 nM LatB. Bars = 5 μm. (**B, C**) Plots of the relative amount of F-actin in pollen grains. The amount of actin filaments was determined by measuring the fluorescence intensity of Alexa-488 phalloidin. The fluorescence intensity of Alexa-488 phalloidin in the absence of LatB was normalized to 100%. The data are presented as mean ± SE ($n = 3$). **$P < 0.01$ by Student's *t* test. Numerical data underlying this panel are available in S7 Data.
(TIF)

**S3 Fig. Overexpression of *CDPK16* renders pollen germination sensitive to the treatment with LatB.** (**A**) Creation of *CDPK16* overexpressors. The level of *CDPK16* transcripts was

determined by qRT-PCR analysis, and the amount of *CDPK16* transcripts in WT was normalized to 1.0. Data are presented as mean ± SE, *n* = 3. Numerical data underlying this panel are available in S8 Data. (**B**) Micrographs of pollen grains and pollen tubes. Pollen derived from WT, *cdpk16* mutants, and *CDPK16* overexpressors were germinated on GM in the presence or absence of 1 nM LatB. Bar = 25 μm. (**C**) Quantification of pollen germination rates. Data are presented as mean ± SE, *n* = 3. ns, no significant difference, *P < 0.05, **P < 0.01 by Student's *t* test. Numerical data underlying this panel are available in S8 Data. (**D**) Quantification of the relative pollen germination rate. Pollen germination rates in the absence of LatB were normalized to 100%. Data are presented as mean ± SE, *n* = 3. *P < 0.05, **P < 0.01 by ANOVA and Student's *t* test. Numerical data underlying this panel are available in S8 Data.
(TIF)

**S4 Fig. Loss of function of *CDPK16* promotes pollen germination and inhibits pollen tube growth.** (**A**) Quantification of pollen germination rate at different time points. Data are presented as mean ± SE, *P < 0.05, **P < 0.01 by Student's *t* test. Numerical data underlying this panel are available in S9 Data. (**B**) Images of pollen tube growth at 2 time points. Single pollen tubes from WT and *cdpk16* mutants were selected for measurement. Bar = 20 μm. (**C**) Quantification of pollen tube growth rate from (B). Data are presented as mean ± SE, **P < 0.01 by Student's *t* test. Numerical data underlying this panel are available in S9 Data.
(TIF)

**S5 Fig. CDPK16 weakly but significantly enhances the actin-depolymerizing activity of ADF4 *in vitro*.** (**A**) SDS-PAGE analysis of the protein samples from a high-speed F-actin co-sedimentation experiment in the presence of $Ca^{2+}$. F-actin, 3 μM; ADF4, 20 μM; CDPK16 (+), 1.0 μM; CDPK16 (++), 2.5 μM; CDPK16 (+++), 5.0 μM. The supernatant fractions (S) and pellets (P) were separated on SDS-PAGE gels, and proteins were detected by Coomassie Brilliant blue R 250 staining. The original pictures are available in S1 Raw Images. (**B**) Quantification of the amount of actin in the supernatant fractions shown in (**A**). Data are presented as mean ± SE, *n* = 3, *P < 0.05 and ns, no significant difference by Student's *t* test. Numerical data underlying this panel are available in S10 Data.
(TIF)

**S6 Fig. Overexpression of *ADF7* alleviates the LatB-resistant pollen germination phenotype in *cdpk16* and loss of function of *CDPK16* and *ADF10* have addictive effect on actin turnover.** (**A**) Micrographs of pollen germinated on GM in the presence or absence of 1.5 nM LatB. Bar = 25 μm. (**B**) Overexpression of *ADF7* alleviates the LatB-resistant pollen germination phenotype in *cdpk16* mutants. The pollen germination rate of different genotypes in the absence of LatB was normalized to 100%. Data are presented as mean ± SE, *n* = 3. **P < 0.01 by Student's *t* test. Numerical data underlying this panel are available in S11 Data. (**C**) Micrographs of pollen germinated on pollen germination medium in the presence or absence of 3 nM LatB. Bar = 25 μm. (**D**) Loss of function of *CDPK16* enhances the LatB-resistant pollen germination phenotype in *adf10* pollen. The pollen germination rate for different genotypes in the absence of LatB was normalized to 100%. Data are presented as mean ± SE, *n* = 3. **P < 0.01 by Student's *t* test. Numerical data underlying this panel are available in S11 Data.
(TIF)

**S7 Fig. Sequence alignment of *Arabidopsis* ADFs.** Protein sequence alignment of 11 *Arabidopsis* ADFs was performed using ESPript3. Green boxes and the black triangle indicate Ser128 in *Arabidopsis* class II ADFs. The predicted secondary structures are indicated above the sequence. The NCBI accession numbers for the sequences are as follows: *Arabidopsis* ADF1 (AtADF1), NP_190187; AtADF2, NP_566882; AtADF3, NP_851227; AtADF4,

NP_851228; AtADF5, NP_565390; AtADF6, NP_565719; AtADF7, NP_194289; AtADF8, NP_567182; AtADF9, NP_195223; AtADF10, NP_568769; AtADF11, NP_171680. (TIF)

**S8 Fig. Sequence alignment of class II ADFs from different plant species.** Protein sequence alignment of class II ADFs from *Arabidopsis* and other plant species was performed using ESPript3. Class II ADFs were selected according to previously published data [71–73]. The green boxes indicate 2 key conserved Serine residues (Ser6 and Ser128) that are associated with ADF function. Ser128 is also indicated by the black triangle. The peptide logo underneath the sequence alignment shows the amino acid prevalence at each position in the peptide sequence encompassing Ser128. The predicted secondary structures are indicated above the sequence. The GenBank accession numbers for ADFs are as follows: AtADF7, NP_194289; AtADF8, NP_567182; AtADF10, NP_568769; AtADF11, NP_171680; *Oryza sativa* ADF1 (OsADF1), Os02g0663800; OsADF6, Os04g0555700; OsADF9, Os07g0484200; *Zea mays* ADF1 (ZmADF1), ACG37280; ZmADF2, NP_001105590; *Solanum lycopersicum* ADF12 (SolycADF12), XP_010317604; SolycADF12-like, XP_015086534; SolycADF7, XP_004240732; SolycADF10-like, XP_019071428; *Brachypodium distachyon* ADF1 (BdADF1), XP_003570044; BdADF6, XP_014751334; BdADF9, XP_010238336; *Populus trichocarpa* ADF3 (PtADF3), XP_002303579; PtADF8, XP_002322471; PtADF10, XP_002318237; POPTR_0001s09170g, XP_002298043; *Cucumis sativus* ADF7 (CsADF7), XP_010448409; KGN57376.1, Csa_3G182120; *Vitis vinifera* ADF1 (VvADF1), XP_010662752; VvADF10, XP_002271495; AMTR_s00140p00036970, XP_006843087; mgv11b015928m; mgv1a015967m; mgv1a026462m; CAA78483.1. (TIF)

**S9 Fig. Lambda protein phosphatase treatment reduces the amount of phosphorylated ADF7 in pollen.** (**A**) The anti-phospho-ADF7(Ser128) antibody specifically recognizes CDPK16-phosphorylated ADF7. The original pictures are available in S1 Raw Images. (**B**, **C**) CDPK16 increases the amount of phosphorylated ADF7 in a dose- and calcium-dependent manner. (**B**) SDS-PAGE analysis of the proteins in the reaction. ADF7, 20 μM; CDPK16, 1.0 μM, 2.5 μM, 5.0 μM, 10 μM. (**C**) Western blot analysis of the protein samples shown in (**B**) probed with anti-phospho-ADF7(Ser128) antibody. The original pictures are available in S1 Raw Images. (**D**) Treatment with λpp (Lambda Protein Phosphatase) reduces the amount of phosphorylated ADF7 in pollen. Total proteins were isolated from mature pollen grains derived from *proADF7::8His-gADF7; adf7*. Western blot analysis of the pulled down 8His-ADF7 probed with anti-ADF7 antibody (left panel, as loading control) and anti-phospho-ADF7(Ser128) antibody (right panel). The original pictures are available in S1 Raw Images. (TIF)

**S10 Fig. CDPK16 enhances the actin-depolymerizing activity of ADF7 but not ADF7^S128A *in vitro*.** (**A**) SDS-PAGE analysis of the protein samples from a high-speed F-actin co-sedimentation experiment in the presence of $Ca^{2+}$. F-actin, 3 μM; ADF7, 20 μM; CDPK16, 4.0 μM. The supernatant fractions (S) and pellets (P) were separated on SDS-PAGE gels, and proteins were detected by Coomassie Brilliant blue R 250 staining. The original pictures are available in S1 Raw Images. (**B**) Quantification of the amount of actin in the supernatant fractions. Data are presented as mean ± SE, $n = 3$, $^*P < 0.05$ and ns, no significant difference by Student's $t$ test. Numerical data underlying this panel are available in S12 Data. (TIF)

**S11 Fig. ADF7^S128A and ADF7^S128D have less activity than ADF7 in rescuing the LatB-resistant pollen tube growth phenotype.** (**A**) qRT-PCR analysis to detect the amount of *ADF7*

transcripts in *adf7 adf10* lines expressing *WT ADF7*, *ADF7*^S128A^, or *ADF7*^S128D^. Numerical data underlying this panel are available in S13 Data. (**B**) Western blot analysis to detect the amount of ADF7 in pollen. Blots were probed with antibodies against UGPase and ADF. The amount of UGPase was used to normalize the amount of ADF7, ADF7^S128A^, and ADF7^S128D^ in pollen. The original pictures are available in S1 Raw Images. (**C**) Quantification of the amount of ADF7 protein in pollen. The relative amount of ADF7 protein from (**B**) is plotted. Data are presented as mean ± SE, $n = 3$, ns, no significant difference, **$P < 0.01$ by Student's *t* test. Numerical data underlying this panel are available in S13 Data. (**D**) Micrographs of pollen tubes. Pollen tubes growing in the absence or presence of 3 nM LatB were presented. Individual pollen tubes at 2 different time points are shown. Bar = 25 μm. (**E**) Quantification of pollen tube growth rate in the absence of LatB. The growth rate of pollen tubes from (**D**, upper panels) is plotted. ns, no significant difference and *$P < 0.05$ by Student's *t* test. Numerical data underlying this panel are available in S13 Data. (**F**) Quantification of relative pollen tube growth rate in the presence of 3 nM LatB. The growth rate of pollen tubes from (**D**, lower panels) is plotted. Data are presented as mean ± SE, $n = 3$, ns, no significant difference, **$P < 0.01$ by Student's *t* test. Numerical data underlying this panel are available in S13 Data.
(TIF)

**S12 Fig. *CDPK16* overexpression renders pollen germination sensitive to LatB treatment for pollen harboring ADF7^S128A^ but not ADF7^S128A^.** (**A**) qRT-PCR analysis to detect the amount of *CDPK16* transcripts in *adf7 adf10* lines expressing ADF7 or ADF7^S128A^. *OECDPK16 8#-6* and *OECDPK16 8#-4* are 2 *CDPK16* overexpressors in the background of *gADF7;adf7 adf10* and *gADF7*^S128A^*;adf7 adf10*, respectively, which have comparable amounts of *CDPK16* transcripts. Numerical data underlying this panel are available in S14 Data. (**B and D**) Micrographs of pollen derived from *gADF7;adf7 adf10*, *gADF7*^S128A^*;adf7 adf10* (**B**), *OECDPK16 8#-6* and *OECDPK16 8#-4* (**D**) in the absence and presence of 3 nM LatB. Scale bar = 100 μm. (**C and E**) Quantification of relative pollen germination rates. Data are the means of 3 replicates ± SE. ns, no significant difference, **$P < 0.01$ (Student's *t* test). Numerical data underlying this panel are available in S14 Data.
(TIF)

**S13 Fig. gCDPK16-eGFP is functional.** (**A**) Micrographs of pollen grains and pollen tubes. Pollen grains derived from WT, *cdpk16-2*, and the complementation line *proCDPK16:: gCDPK16-eGFP; cdpk16-2* were germinated on the surface of GM for 3 h in the presence or absence of 1.5 nM LatB. Bar = 25 μm. (**B**) Quantification of relative pollen germination rate. Pollen germination rate in the absence of LatB was normalized to 100%. Data are presented as mean ± SE, $n \geq 500$, **$P < 0.01$ by Student's *t* test. The experiments were repeated 3 times. Numerical data underlying this panel are available in S15 Data.
(TIF)

**S14 Fig. CKL2 enhances the actin-depolymerizing activity of ADF7 *in vitro*.** (**A**) SDS-PAGE analysis of the protein samples from a high-speed F-actin co-sedimentation experiment in the presence of Ca$^{2+}$. F-actin, 3 μM; ADF7, 20 μM; CKL2 (+), 1.0 μM; CKL2 (++), 2.5 μM; CKL2 (+++), 5.0 μM. The supernatant fractions (S) and pellets (P) were separated on SDS-PAGE gels, and proteins were detected by Coomassie Brilliant blue R 250 staining. The original pictures are available in S1 Raw Images. (**B**) Quantification of the amount of actin in the supernatant fractions shown in (**A**). Data are presented as mean ± SE, $n = 3$, **$P < 0.01$ and ns, no significant difference by Student's *t* test. Numerical data underlying this panel are available in S16 Data.
(TIF)

**S1 Table. Primers used in this study.**
(DOCX)

**S1 Data. Source data underlying Fig 1B.**
(XLSX)

**S2 Data. Source data underlying Fig 2A, Fig 2B and 2C.**
(XLSX)

**S3 Data. Source data underlying Fig 3B, 3D, 3F and 3H.**
(XLSX)

**S4 Data. Source data underlying Fig 4A, 4B, 4D, 4G, 4H and 4J.**
(XLSX)

**S5 Data. Source data underlying Fig 5B–5F.**
(XLSX)

**S6 Data. Source data underlying S1B, S1E and S1G Fig.**
(XLSX)

**S7 Data. Source data underlying S2B and S2C Fig.**
(XLSX)

**S8 Data. Source data underlying S3A, S3C and S3D Fig.**
(XLSX)

**S9 Data. Source data underlying S4A and S4C Fig.**
(XLSX)

**S10 Data. Source data underlying S5B Fig.**
(XLSX)

**S11 Data. Source data underlying S6B and S6D Fig.**
(XLSX)

**S12 Data. Source data underlying S10B Fig.**
(XLSX)

**S13 Data. Source data underlying S11A, S11C, S11E and S11F Fig.**
(XLSX)

**S14 Data. Source data underlying S12A, S12C and S12E Fig.**
(XLSX)

**S15 Data. Source data underlying S13B Fig.**
(XLSX)

**S16 Data. Source data underlying S14B Fig.**
(XLSX)

**S1 Raw Images. Unprocessed images of all gels and blots in the paper.**
(PDF)

**S1 Movie. Dynamics of actin filaments decorated with Lifeact-eGFP in a growing WT pollen tube.** Movie corresponds to time-lapse images of the WT pollen tube shown in the upper panel of Fig 1A. Images were captured every 3 s and displayed at 5 frames per second in the

movie. Bar = 10 μm.
(AVI)

**S2 Movie. Dynamics of actin filaments decorated with Lifeact-eGFP in a growing *cdpk16-1* pollen tube.** Movie corresponds to time-lapse images of the *cdpk16-1* pollen tube shown in the lower panel of Fig 1A. Images were captured every 3 s and displayed at 5 frames per second in the movie. Bar = 10 μm.
(AVI)

**S3 Movie. Dynamics of apical actin filaments in a growing WT pollen tube.** Movie corresponds to time-lapse images of actin filaments decorated with Lifeact-eGFP within the apical region of the WT pollen tube shown in the left panel of Fig 1C. Images were captured every 3 s and displayed at 2 frames per second in the movie. Bar = 5 μm.
(AVI)

**S4 Movie. Dynamics of apical actin filaments in a growing *cdpk16-1* pollen tube.** Movie corresponds to time-lapse images of actin filaments decorated with Lifeact-eGFP within the apical region of the *cdpk16-1* pollen tube shown in the right panel of Fig 1C. Images were captured every 3 s and displayed at 2 frames per second in the movie. Bar = 5 μm.
(AVI)

**S5 Movie. Dynamics of actin filaments *in vitro* visualized by TIRFM.** Movie corresponds to time-lapse images of actin alone, as shown in Fig 3G. Images were captured every 3 s and displayed at 5 frames per second in the movie. Bar = 10 μm.
(AVI)

**S6 Movie. Dynamics of actin filaments *in vitro* in the presence of CDPK16 visualized by TIRFM.** Movie corresponds to time-lapse images of actin filaments in the presence of 125 nM CDPK16, as shown in Fig 3G. Images were captured every 3 s and displayed at 5 frames per second in the movie. Bar = 10 μm.
(AVI)

**S7 Movie. Dynamics of actin filaments *in vitro* in the presence of ADF7 visualized by TIRFM.** Movie corresponds to time-lapse images of actin filaments in the presence of 500 nM ADF7, as shown in Fig 3G. Images were captured every 3 s and displayed at 5 frames per second in the movie. White arrows indicate severing events of actin filaments. Bar = 10 μm.
(AVI)

**S8 Movie. Dynamics of actin filaments *in vitro* in the presence of ADF7 and CDPK16 visualized by TIRFM.** Movie corresponds to time-lapse images of actin filaments in the presence of 500 nM ADF7 and 125 nM CDPK16, as shown in Fig 3G. Images were captured every 3 s and displayed at 5 frames per second in the movie. White arrows indicate severing events of actin filaments. Bar = 10 μm.
(AVI)

**S9 Movie. Dynamics of CDPK16-eGFP during pollen germination.** Movie corresponds to time-lapse images of the *gCDPK16-eGFP;cdpk16-2* pollen grains shown in Fig 6A. Images were captured every 30 s and displayed at 5 frames per second in the movie. White arrows indicate growth direction. Bar = 10 μm.
(AVI)

**S10 Movie. Dynamics of CDPK16-eGFP during pollen tube growth.** Movie corresponds to time-lapse images of the *gCDPK16-eGFP;cdpk16-2* pollen tube shown in Fig 6B. Images were

captured every 3 s and displayed at 12 frames per second in the movie. Bar = 10 μm. (AVI)

## Acknowledgments

We thank Prof. Qijun Chen (China Agricultural University) for providing the CRISPR/Cas9 vector. We also thank Ms. Yingchao Li (Institute of Botany, Chinese Academy of Sciences) for the screening of T-DNA insertion mutants whose pollen germination is resistant to LatB. We are also grateful to Xianbin Meng at the Center of Biomedical Analysis, Tsinghua University, for the analysis of LC-MS/MS data.

## Author Contributions

**Conceptualization:** Shanjin Huang.

**Data curation:** Qiannan Wang, Yanan Xu, Shuangshuang Zhao, Yan Guo.

**Funding acquisition:** Shanjin Huang.

**Investigation:** Qiannan Wang, Yanan Xu, Shuangshuang Zhao, Yuxiang Jiang, Ran Yi.

**Methodology:** Qiannan Wang, Yanan Xu, Shuangshuang Zhao, Yuxiang Jiang.

**Project administration:** Shanjin Huang.

**Resources:** Yan Guo.

**Software:** Qiannan Wang, Yanan Xu.

**Supervision:** Qiannan Wang, Shanjin Huang.

**Validation:** Yanan Xu, Ran Yi, Yan Guo.

**Visualization:** Qiannan Wang, Yanan Xu.

**Writing – original draft:** Qiannan Wang, Yanan Xu, Shanjin Huang.

**Writing – review & editing:** Shanjin Huang.

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
