## [Editor Report · Decision Letter 0]

28 Sep 2021

Dear Dr Huang, 

Thank you for submitting your manuscript entitled "Phosphorylation-mediated Activation of Actin-depolymerizing Factor in Arabidopsis" for consideration as a Research Article by PLOS Biology.

Your manuscript has now been evaluated by the PLOS Biology editorial staff as well as by an academic editor with relevant expertise and I am writing to let you know that we would like to send your submission out for external peer review.

Please re-submit your manuscript within two working days, i.e. by Sep 30 2021 11:59PM.

Kind regards,

Ines

--

Ines Alvarez-Garcia, PhD

Senior Editor

PLOS Biology

---

## [Decision Letter · Decision Letter 1]

25 Nov 2021

Dear Dr Huang,

Thank you for submitting your manuscript entitled "Phosphorylation-mediated Activation of Actin-depolymerizing Factor in Arabidopsis" for consideration as a Research Article at PLOS Biology. Thank you also for your patience as we completed our editorial process, and please accept my apologies for the delay in providing you with our decision. Your manuscript has been evaluated by the PLOS Biology editors, an Academic Editor with relevant expertise, and by three independent reviewers.

You will see that the three reviewers are positive and think that the findings of the manuscript are significant for the field. Nevertheless, they all recommend several additional experiments that are needed to strengthen the conclusions. After consulting with the Academic Editor and the rest of the editorial team, we would like to invite you to submit a revision that addresses all the issues raised by the reviewer except for the gel filtration profiles of the proteins used in the in vitro analyses, which we won’t make a requirement for publication.

In light of the reviews (attached below), we will not be able to accept the current version of the manuscript, but we would welcome re-submission of a revised version that takes into account the reviewers' comments. We cannot make any decision about publication until we have seen the revised manuscript and your response to the reviewers' comments. Your revised manuscript is also likely to be sent for further evaluation by the reviewers.

We expect to receive your revised manuscript within 3 months. 

**IMPORTANT - SUBMITTING YOUR REVISION**

3. Resubmission Checklist

a) *PLOS Data Policy*

b) *Published Peer Review*

d) *Blurb*

Please also provide a blurb which (if accepted) will be included in our weekly and monthly Electronic Table of Contents, sent out to readers of PLOS Biology, and may be used to promote your article in social media. The blurb should be about 30-40 words long and is subject to editorial changes. It should, without exaggeration, entice people to read your manuscript. It should not be redundant with the title and should not contain acronyms or abbreviations. For examples, view our author guidelines: https://journals.plos.org/plosbiology/s/revising-your-manuscript#loc-blurb

Sincerely,

Ines

--

Ines Alvarez-Garcia, PhD

Senior Editor

PLOS Biology

Reviewers' comments

Rev. 1:

This manuscript entitled "Phosphorylation-mediated Activation of Actin-depolymerizing Factor in Arabidopsis" by Wang et al provided a molecular link between Ca2+/CDPK signaling and actin cytoskeleton organization in pollen tubes. Previous studies have found that plant ADFs are phosphorylated and inactivated by CDPK(s), whereas this paper reveals that CDPK16-mediated phosphorylation upregulates ADF7's activity in promoting actin depolymerizing and severing at the apical and subapical regions of pollen tubes.

Through providing substantial evidence both in vivo and in vitro, the authors enhance our understanding of the regulation of ADF in plant community. This paper is well-organized and falls within the scope of PLOS Biology.

To further improve the manuscript, I have listed some remarks.

1. The authors said that they performed a forward chemical genetic screening for mutants showing alteration in LatB sensitivity and found one mutant allele of CDPK16. More details should be added for how the gene being identified.

2. The subhead "Overexpression of ADF7 Alleviates the Actin Turnover Defects in cdpk16 Pollen, and Loss of Function of CDPK16 Enhances the Actin Turnover Defects in adf10 Pollen". The phenotype they provided in Fig S5 is only about pollen germination, so the word "Actin Turnover Defects" is not accurate. This also exist in other parts of the manuscript (e.g. Fig S9), please confirm and modify them.

3. In Fig 5A, they showed that actin filaments were brighter in proADF7::gADF7S128A; adf7 adf10 pollen tubes than in adf10 and proADF7::gADF7; adf7 adf10 pollen tubes. However, the in vitro assays in Fig 4 found that ADF7S128A had roughly similar activity to ADF7. How to explain the difference between the function of ADF7S128A in vivo and in vitro? Moreover, I noticed that they have the plant material of proADF7::gADF7S128D; adf7 adf10. And the actin filament organization of this plant material also need to be displayed and quantified in Fig 5 to better support the in vitro evidence.

4. In addition, it worth to check if CDPK16-eGFP and ADF7 show some co-localizations in pollen tubes. This will provide another evidence for the functional linkage of CDPK16 and ADF7.

Rev. 2:

The current manuscript by Wang et al is the second in a developing series that describes the identification of a kinases which regulate key actin binding proteins (in this case, ADF), required for the modulation and regulation of actin cytoskeletal dynamics. In the current study, the authors demonstrate that CDPK16 interacts with ADF7, and that this interaction/activity enhances ADF7-mediated actin depolymerization and severing. Using a genetics-based approach, they observed that the rate of actin turnover is reduced in cdpk16 mutant pollen. CDPK16 phosphorylates ADF7 at Ser-128.

At an experimental level, the current manuscript is sound; it describes previously unknown activities of CDPK16 and ADF. At a broader, fundamental level, the current study builds upon known mechanisms and activities required for the regulation of ADF activity.

Based on previous work by the Huang lab, as well as those of others, I feel it would be necessary to delve into the specificity of this interaction a bit more. Previous work from the Huang lab identified a kinase required for the regulation of ADF4 activity. Does CDPK16 p-late ADF4? I recognize that they tested ADF10, which is also expressed in pollen tubes. I believe it would be useful to evaluate the kinase interaction activity of a third ADF - one that is not expressed in pollen.

Related to the above, the Huang group evaluated CKL2 (Plant Cell, 2016). Does CKL2 p-late ADF7? I do not mention these recommended experiments for the sake of adding additional tasks, yet to express that the authors, and others, are developing models which demonstrate a mechanism that describes specificity of ADF activity through phospho-regulation and residue specificity. It is important, that as this develops, to ensure a stringent analysis.

What is the biological consequence of, for example, a cdpk6 mutant, or a adf7/Ser-128 plant? Are there reproductive consequences of these isoforms? I fully recognize that the current study describes an impact on the individual filament dynamics, but what is the "next step"? What is the impact on the biology of the cell, the tissue, or the plant on these modifications?

Rev. 3:

In their manuscript, Wang et al. propose a change in the current regulatory paradigm of ADF factors, regarding their actin depolymerizing and severing activity. The authors propose that ADF proteins (particularly ADF7) get phosphorylated in position S128, activating their depolymerizing and severing function. The authors identify a CDPK protein (CDPK16) to be involved in this regulatory process.

The paper is well written, and the authors combine in vivo and in vitro approaches to shed light on this regulatory process. However there would be interesting questions to address:

Major:

- The authors describe that cdpk16 knock-out mutant has a reduced rate of acting turnover, by accumulating actin filaments. Next, The authors connect CDPK16 to ADF7 in vivo and in vitro, by Luciferase interaction assays and through enrichment of CDPK16 hits when pulling down ADF7. Suggesting CDPK16 to be an important regulator of the phosphorylated status of ADF7. However, the authors seem to be able to overcome the cdpk16 phenotype by overexpressing ADF7. If ADF7 needs to be phosphorylated to activate is depolymerization function, How do the authors explain this data? Are there other cdpks in the tissue that could replace cdpk16?

- To see the in vivo relevance of CDPK16 in phosphorylating ADF7 in the Ser128 residue, the authors should check how much of ADF7 S128 phosphorylated they find with their specific ADF7-Ser antibody in a cdpk16 mutant vs wt. Is the pool of non-phosphorylated vs phosphorylated protein substantially different? Alternatively, if the antibody would not be sensitive enough they could do an MS comparison between ADF7 in the cdpl16 vs Wild-type background.

- The authors try to express a phosphomimic version of ADF7, S128D in planta to analyze its role in actin turnover. The mutant does not behave as an activator but rather the contrary as expected, which, as the authors suggests, could be due to the fact that the change to Aspartate would not fully mimic the function of a phosphorylated residue. To prove the role active role of ADFS128 phosphorylated, the authors could overexpress CDPK16 with a strong pollen specific promoter (pACA9) into their backgrounds pADF7:ADF7S128A and pADF7:ADF7 the adf7adf10 background. In this case they should increase the pool of phosphorylated ADF7S128 and see its enhanced effect in actin depolymerization.

- The authors should also include gel filtration profiles of the proteins they use in their in vitro analyses in the supplementary data, of wild type and mutants. To ensure the proteins are properly folded, especially in the case of the mutants used in their different in vitro assays.

Minor:

In figure S8 the authors forgot to include marker in the figures C and D. Also in figure C is a bit confusing to know which lanes contain ADF7, maybe that could be indicated.

---

## [Decision Letter · Decision Letter 2]

28 Feb 2023

Dear Dr Huang,

Thank you for your patience while we considered your revised manuscript entitled "Phosphorylation-mediated Activation of Actin-depolymerizing Factor in Arabidopsis" for publication as a Research Article at PLOS Biology. This revised version of your manuscript has been evaluated by the PLOS Biology editors, the Academic Editor and the three original reviewers.

Based on the reviews (attached below), we are likely to accept this manuscript for publication, provided you satisfactorily address the remaining points raised by Reviewer 1. Please also make sure to address the data and other policy-related requests stated below.

In addition, and following one of Rev. 1's requests, we would like you to consider a suggestion to improve the title:

"Activation of actin-depolymerizing factor by CDPK16-mediated phosphorylation promotes actin turnover in Arabidopsis pollen tubes"

We expect to receive your revised manuscript within two weeks. 

*Published Peer Review History*

*Press*

Sincerely,

Ines

--

Ines Alvarez-Garcia, PhD

Senior Editor

PLOS Biology

DATA POLICY:

Fig. 1B; Fig. 2B, C; Fig. 3B, D, F, H; Fig. 4A, B, D, G, H, J; Fig. 5B-F; Fig. S1B, E, G; Fig. S2B, C; Fig. S3A, C, D; Fig. S4A, C; Fig. S5B; Fig. S6B, D; Fig. S10B; Fig. S11A, C, E, F; Fig. S12A, C, E; Fig. S13B and Fig S14B

We require the original, uncropped and minimally adjusted images supporting all blot and gel results reported in an article's figures or Supporting Information files. We will require these files before a manuscript can be accepted so please prepare and upload them now. Please carefully read our guidelines for how to prepare and upload this data: https://journals.plos.org/plosbiology/s/figures#loc-blot-and-gel-reporting-requirements

BLURB

Please also provide a blurb which (if accepted) will be included in our weekly and monthly Electronic Table of Contents, sent out to readers of PLOS Biology, and may be used to promote your article in social media. The blurb should be about 30-40 words long and is subject to editorial changes. It should, without exaggeration, entice people to read your manuscript. It should not be redundant with the title and should not contain acronyms or abbreviations. For examples, view our author guidelines: https://journals.plos.org/plosbiology/s/revising-your-manuscript#loc-blurb

Reviewers' comments

Rev. 1:

The authors have answered most of the questions I raised, and the new data they added in the revised manuscript provide more evidence to sustain their finding. But, still I have another two minor questions for the authors to address:

1. The title of the manuscript is too brief, which did not include the main findings. The specific function and biological significance of CDPK16-mediated phosphorylation of ADF7 is suggested to be highlighted in the title and in abstract.

2. The main point of the manuscript is that CDPK16-mediated phosphorylation upregulates ADF7 and most of the data supports this argument. However, it's somewhat inconsistent when it comes to the phosphorylation site S128 of ADF7. "ADF7S128D has reduced capability in promoting actin turnover in pollen compared to ADF7". This is a confusing part of the manuscript. The authors simply explain that ADF7S128D cannot fully mimic the function of phosphorylated ADF7 in pollen. Could the author give more specific explanation or speculation? The possible reasons need to be carefully analyzed and added in the discussion section. Is it possible that the other phosphorylation sites of ADF7 is involved?

Rev. 2:

In the second revision of this manuscript, I would like to acknowledge that the authors have addressed not only my primary concerns (in both rounds of review), but have addressed the major (and minor) elements noted by the 2 additional reviewers. As noted in my previous reviews, this is a sound manuscript and builds upon current developing work in the field of actin biology. I have no additional comments, and I believe that this manuscript will be a substantial contribution to the field.

Rev. 3:

The authors have addressed the majority of the comments improving the manuscript.

---

## [Editor Report · Decision Letter 3]

11 Mar 2023

Dear Dr Huang,

Thank you for the submission of your revised Research Article entitled "Activation of Actin-depolymerizing Factor by CDPK16-mediated Phosphorylation Promotes Actin Turnover in Arabidopsis Pollen Tubes" for publication in PLOS Biology. On behalf of my colleagues and the Academic Editor, Mark Estelle, I am delighted to say that we can in principle accept your manuscript for publication, provided you address any remaining formatting and reporting issues. These will be detailed in an email you should receive within 2-3 business days from our colleagues in the journal operations team; no action is required from you until then. Please note that we will not be able to formally accept your manuscript and schedule it for publication until you have completed any requested changes.

PRESS

Sincerely, 

Ines

--

Ines Alvarez-Garcia, PhD

Senior Editor

PLOS Biology
